# Targeting Nuclear Receptors in Lung Cancer—Novel Therapeutic Prospects

**DOI:** 10.3390/ph15050624

**Published:** 2022-05-18

**Authors:** Shailendra Kumar Gangwar, Aviral Kumar, Kenneth Chun-Hong Yap, Sandra Jose, Dey Parama, Gautam Sethi, Alan Prem Kumar, Ajaikumar B. Kunnumakkara

**Affiliations:** 1Cancer Biology Laboratory, Department of Biosciences and Bioengineering, Indian Institute of Technology (IIT) Guwahati, Guwahati 781039, India; shailendra00780@gmail.com (S.K.G.); aviralkmr@gmail.com (A.K.); sandrajose033@gmail.com (S.J.); piudey888@gmail.com (D.P.); 2Department of Pharmacology, Yong Loo Lin School of Medicine, National University of Singapore, Singapore 117600, Singapore; yapchkenneth@gmail.com (K.C.-H.Y.); phcgs@nus.edu.sg (G.S.); 3NUS Centre for Cancer Research, Yong Loo Lin School of Medicine, National University of Singapore, Singapore 117597, Singapore

**Keywords:** lung cancer, nuclear receptors, agonists/antagonists, biomarker, cell growth

## Abstract

Lung cancer, the second most commonly diagnosed cancer, is the major cause of fatalities worldwide for both men and women, with an estimated 2.2 million new incidences and 1.8 million deaths, according to GLOBOCAN 2020. Although various risk factors for lung cancer pathogenesis have been reported, controlling smoking alone has a significant value as a preventive measure. In spite of decades of extensive research, mechanistic cues and targets need to be profoundly explored to develop potential diagnostics, treatments, and reliable therapies for this disease. Nuclear receptors (NRs) function as transcription factors that control diverse biological processes such as cell growth, differentiation, development, and metabolism. The aberrant expression of NRs has been involved in a variety of disorders, including cancer. Deregulation of distinct NRs in lung cancer has been associated with numerous events, including mutations, epigenetic modifications, and different signaling cascades. Substantial efforts have been made to develop several small molecules as agonists or antagonists directed to target specific NRs for inhibiting tumor cell growth, migration, and invasion and inducing apoptosis in lung cancer, which makes NRs promising candidates for reliable lung cancer therapeutics. The current work focuses on the importance of various NRs in the development and progression of lung cancer and highlights the different small molecules (e.g., agonist or antagonist) that influence NR expression, with the goal of establishing them as viable therapeutics to combat lung cancer.

## 1. Introduction

Lung cancer is the major causative factor for cancer-related deaths globally both for men and women. According to the GLOBOCAN 2020, lung cancer incidence was reported to be 2.2 million new cases (11.4% of all sites) and 1.8 million deaths (18% of all sites), making it the second most commonly diagnosed cancer worldwide [1]. The highly heterogeneous nature of lung cancer and its varying symptoms and signs can be stratified into two broad classes differing in their sites of growth and spread: small-cell lung cancer (SCLC) and non-small-cell lung cancer (NSCLC). SCLC accounts for 10–15% of all lung cancers and tends to be highly aggressive with metastasis into submucosal lymphatic vessels and lymph nodes. The NSCLCs are further subdivided into squamous cell lung cancers (SCCs) arising in the main bronchi and representing about 25–30% of all lung cancers, adenocarcinomas (ADKs), constituting around 40% and arising in peripheral bronchi, and large-cell carcinoma (LCC), which account for 10%, and its cancer cells are not observed with classic glandular or squamous morphology [2]. Transformation of a normal cell to a malignant state consists of a multistep process involving a series of alterations at the genetic and epigenetic level, which then advances cancer into an invasive form through clonal expansion. Development of a primary tumor is followed by additional mutations in the signaling cascades during clonal expansion, leading to invasion, metastasis, and resistance to therapy [3,4]. In order to develop diagnostics, and reliable therapies, these changes occurring overtime in a lung cell need to be identified and characterized [5,6]. Cigarette smoking is the main risk factor for the development of lung tumor with 80–90% cases observed in smokers. Most of the lung cancers are observed in smokers with around 85% of NSCLC and 98% of SCLC cases. Tobacco smoking is the major risk factor for lung cancer, as tobacco contains over 20 carcinogenic compounds, including polycyclic aromatic hydrocarbons and nicotine-derived nitrosoaminoketone, which can destabilize DNA by inducing mutations through DNA adducts [7,8]. Other risk factors responsible for lung cancer involve genetics, such as family history, high-penetrance genes and genetic polymorphism, alcohol, and diet such as high consumption of meat [9,10]. The risk for lung cancer is higher for those who have chronic inflammation resulting from infections and other medical complications such as chronic obstructive pulmonary disease and pulmonary tuberculosis [11,12,13,14]. Ionizing radiation exposure also increases the chances of lung cancer, which has been reported in atomic bomb survivors and patients treated with radiation therapy [15]. Hence, a further understanding of the complex etiologies and deregulation at the molecular level of lung cancer is imperative for the development of novel therapeutic regimens circumventing this disease.

Nuclear receptors (NRs) are transcription factors (TFs) involved in a plethora of biological activities, such as embryogenesis, differentiation, homeostasis, metabolism, immunity, and cell proliferation [16,17,18,19,20]. There are 48 human NRs, including receptors for a range of ligands, such as steroid hormones, thyroid hormones, retinoic acid, vitamin D, fatty acids, and oxysterols. NRs have been extensively studied and researched to uncover how different disorders, including cancer and inflammatory diseases, may be regulated in a transcriptional manner, as well as their normal and physiological states [21,22,23,24,25,26,27,28]. The conformational shift induced by ligand interaction causes NRs to bind to a specific DNA sequence (called nuclear receptor response elements (NRREs)) located all across the genome, which aids in the expression of numerous target genes by engaging many coregulators and coactivators. NRs are divided into three categories depending on their binding with the ligands: endocrine, adopted, and orphan NRs. Most of the tissues in the body express numerous types of NRs that are sensitive to diverse types of ligands, and they function in the control of the expression of target genes required in tumor formation and suppression [29]. NRs play a crucial role in cellular activities as well as in a diverse range of pathological diseases, including cancer [21,30,31,32,33]. One of the earliest investigations found a link between steroid hormones and prostate cancer, and growing evidence indicates that NRs play a vital role in cancer development and progression [34,35,36,37,38]. The NR superfamily is implicated in cancer initiation and progression as oncogenic signaling is dependent on NR-mediated transcriptional control. The expression of different NRs has been well-explored in a variety of cancers, and they were shown to have either oncogenic or tumor-suppressive functions. The NRs associated with cancers include the retinoic acid receptors (RARs) [39,40,41], the retinoid X receptors (RXRs) [41,42,43], the peroxisome-proliferator-activated receptor γ (PPARγ) [43,44,45,46], and the vitamin D receptor (VDR) [47,48,49,50,51]. Because of their physiological activation by low-molecular-mass ligands, many NRs have been considered tractable small-molecule therapeutic targets through significant investigations [52,53,54]. With the advent of discovering small compounds targeting these receptors [55,56,57,58,59], substantial progress has been achieved in the treatment and prevention of estrogen receptor (ER)^+^ breast cancer and castration-resistant prostate cancer (CRPC) [60,61,62,63]. It is well-known that NRs modulate the expression of various hormonal signaling and cancers dependent on such pathways can be targeted using natural or modified agonists and antagonists. Several clinical and preclinical trials are now being conducted to assess the therapeutic efficacy of ER and AR inhibitors in the treatment of cancer, either as monotherapy or combined therapy. Various other NRs have been investigated as potential therapeutic targets in cancer, which include the glucocorticoid receptor (GR), progesterone receptor (PR), RARs, and RXRs [64,65]. The relevance of NRs in lung cancer has been demonstrated by the ability of agonists and antagonists to inhibit tumor development, invasion, and migration, and cause induction of apoptosis by regulating a large number of genes involved in cell proliferation, differentiation, and death [66,67].

Thus, the present study focuses on the crucial role of various NRs in the development and progression of both SCLC and NSCLC (Figure 1). It also highlights how a particular small molecule (e.g., agonist or antagonist) affects the expression of various types of NRs crucial for cell growth and maintenance with the purpose of establishing them as potential therapeutics against lung cancer.

## 2. NRs in Lung Cancer

NRs are well-documented to regulate the growth of cancers in hormone-driven tissues, compared to other cancers (Figure 2). The importance of NRs in NSCLC/SCLC is fascinating because the lung is not considered as an organ driven by hormones. A plethora of studies have implicated the role of NRs in lung cancer and that their expression might be deregulated in this disease (Table 1). In addition, mRNA levels of NRs can be useful as a prognostic factor for patient outcomes [68]. They have different roles in modulating several hallmarks of tumorigenic processes, such as cell growth, differentiation, homeostasis, migration, invasion, and cell apoptosis, thereby exerting crucial effects on tumor behavior (Table 2).

### 2.1. Androgen Receptors (ARs)

AR is a ligand-dependent TF belonging to the NR superfamily. When bound to androgen, it forms a homodimer that binds to the promoter region of target genes, thereby regulating many cellular activities, such as growth, differentiation, and survival, in AR-expressing cells. AR consists of three functional domains: an N-terminal transcriptional regulatory domain, which is the most variable part of the protein, a DNA-binding domain (DBD), and a ligand-binding domain (LBD) (Figure 3B). In the AR, the DBD is the most conserved region among the other members of the steroid hormone receptor family. The ligand-independent activation function-1 (AF-1) found in the N-terminal domain (NTD) is necessary for AR to function at its highest level [69]. The LBD contains AF-2, which is required for the formation of coregulator binding sites and interactions between the NTD and LBD [70,71]. The DBD and LBD are linked by a hinge region. Two zinc-finger domains in the DBD attach to specific DNA regions and are responsible for the direct binding of AR to the promoter and other areas of target genes, followed by activation or repression of genes [72] (Figure 3A).

The expression of AR is observed to be altered in lung cancerous tissues [73]. Various studies have investigated the role of AR, suggesting its importance in the initiation and advancement of lung cancer. One such study showed that athymic nude mice transplanted with eight different lung tumor cell lines demonstrated low AR content in all analyzed tumor cell lines [74]. Another report demonstrated that NSCLC cell lines expressed AR at low levels, whereas SCLC cell lines had no AR expression [75]. In contrast to that, a study conducted on female C57BL/6 mice, transplanted tumor and lung tissues were shown to have higher expression of ARs and other reproductive hormone receptors [76]. Maasberg and colleagues showed that treatment with testosterone, an AR agonist, resulted in growth stimulation up to 3-fold in five different lung cancer cell lines examined with positive AR expression. When AR antagonists such as flutamide and cyproterone acetate were used to treat, they antagonized the growth stimulatory effects of testosterone, suggesting that testosterone possesses an ability to stimulate the growth of lung cancer cells [77]. Another AR agonist, 5-alpha-dihydrotestosterone (DHT), was reported to significantly increase the cell growth in the H1355 lung adenocarcinoma cell line [78]. A different study on DHT observed that it stimulated cell growth in A549 cell lines by upregulating cyclinD1 (CD1). Moreover, this study suggests that the activation of the mammalian target of rapamycin (mTOR)/CD1 pathway by p38 mitogen-activated protein kinase (MAPK) might be the approach for AR and epidermal growth factor receptor (EGFR) cross-talk, leading to lung cancer development [79]. As a result of studies conducted over the years, it is suggested that ARs can promote cell growth by binding to their agonist, and suppressing ARs in tumors might be a viable approach in abrogating lung cancer cell proliferation and its progression.

**Figure 3 pharmaceuticals-15-00624-f003:**
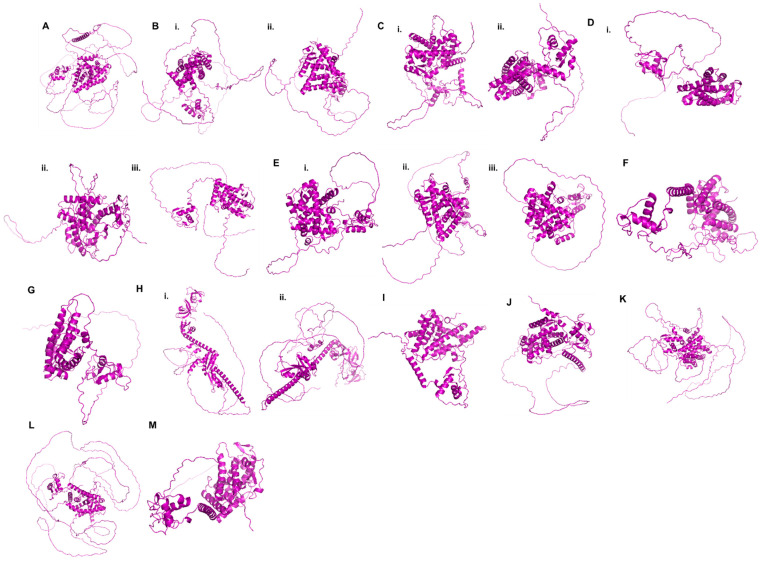
**Structural representation of different NRs involved in lung cancer**: (**A**) AR (Uniprot ID: P10275), (**B**) i. ERα (Uniprot ID: P03372), ii. ERβ (Uniprot ID: Q92731), (**C**) i. PPARγ (Uniprot ID: Q03181), ii. PPARδ (Uniprot ID: P37231), (**D**) i. RARα (Uniprot ID: P10276), ii. RARβ (Uniprot ID: P10826), iii. RARγ (Uniprot ID: P13631), (**E**) i. RXRα (Uniprot ID: P19793), ii. RXRβ (Uniprot ID: P28702), iii. RXRγ (Uniprot ID: P48443), (**F**) VDR (Uniprot ID: P11473), (**G**) ERRα (Uniprot ID: P11474), (**H**) i. FXR1 (Uniprot ID: P51114), ii. FXR2 (Uniprot ID: P51116), (**I**) PXR (Uniprot ID: O75469), (**J**) LRH1 (Uniprot ID: O00482), (**K**) GR (Uniprot ID: P04150), (**L**) PR (Uniprot ID: P06401), (**M**) TRα (Uniprot ID: P10827). The primary structures of these proteins were downloaded from the UniProt. The structures were predicted for these proteins using AlphaFold protein structure database. AlphaFold utilizes artificial intelligence to predict a protein’s three-dimensional structure from the given amino acid sequence [80,81,82].

### 2.2. Estrogen Receptors (ERs)

The ER is an NR for the steroid hormone estrogen, which modulates the expression of different genes, playing a role in cell proliferation and survival. When attached to a ligand, phosphorylation of the ER occurs with the dissociation from the heat shock proteins followed by dimerization. The receptor is then translocated to the nucleus and binds to specific DNA sequences known as estrogen-responsive elements (EREs) found in the promoters of target genes recruiting coactivators and corepressors, relaying a range of responses and influencing physiological functions [83,84,85]. There are two ER subtypes: ERα and ERβ, which are encoded by different genes and found in a wide variety of organs. ERs have been known to contain six separate functional domains in their structure (Figure 3B) [86,87,88,89]. Despite their structural differences, the N-terminal A/B domains have 17% amino acid homology. With 97% similarity, the DBD is found in the C region. The D domain, having 36% homology, also known as the hinge domain, has a nuclear localization signal (NLS) and links the D region to the carboxyl terminal (E) domain (CTD). The LBD is the E domain, which has 56% amino acid homology. It has a site for hormone binding, a dimerization interface for receptors (homo- and heterodimerization), and a ligand-dependent AF-2. The F domain is present in the extreme carboxyl terminus and shares 18% homology [83].

In normal cells, ER-mediated signaling is required for various physiological tasks, such as growth, differentiation, and cell death [90,91,92]. It is well-known that ER-mediated signals have a profound impact on the development and progression of many malignancies. Estrogens can function as direct carcinogens by converting to catechol estrogens and forming DNA adducts, and they might increase tumor growth via a receptor-mediated signaling mechanism [93,94]. The presence of ERs in lung cancers has been reported to be conflicting in several studies [75,95,96,97,98,99,100]. A couple of in vitro and in vivo studies showed very low expression of estrogen receptors in NSCLC cell lines [74,75]. Interestingly, a report on 44 lung cancer patients found 16% expression of ERs in tumor samples [101]. In contrast, many studies have shown ERs being overexpressed in lung cancer tissues [95,102,103]. There were no significant links between ER content and age, blood type, histological assessment, tumor size, or regional lymph node metastases; however, there were significant associations between ER content and lung cancer prognosis [95,102,103]. Therefore, antiestrogen treatment may be possible for individuals with advanced bronchogenic carcinoma with improved discovery and characterization of ERs. In line with this, ERα and ERβ immunohistochemistry expression in 317 NSCLCs samples showed an increased expression of both receptors, suggesting that the expression of ERα and ERβ identifies a specific subpopulation of NSCLC with distinct clinicopathologic and genetic characteristics [104]. Another study demonstrated higher and lower expression of ERα and ERβ, respectively, in resected NSCLC specimens from the patients, serving as a poor prognosis in NSCLC patients, and the absence of ERβ might be used as a plausible marker to identify individuals who are at higher risk in their treatment [105]. In another study, cell lines, such as 91T, 784T, and 54T, were observed to have high levels of ER mRNA, whereas the 128-88T, H23, A549, and Calu-6 cell lines reported low levels, suggesting that the ER is not only confined in ADKs, but also present in SCCs [106].

Treatment with the agonist, such as beta-estradiol, was shown to stimulate proliferation in the NSCLC line, H23, and SCID xenograft mice. ERβ immunoreactivity was found in the nucleus of NSCLCs, but ERα immunoreactivity was significant in the cytoplasm, implying that both nuclear and cytoplasmic signaling may be implicated in estrogenic responses in the lung tumorigenesis [106]. While studying the expression of both ER subtypes, it was demonstrated that there were 87 (38.2%) ERα-positive NSCLCs and 77 (33.8%) ERβ-positive NSCLCs among the 228 patients [107]. Overexpressed ERα in NSCLC patients was correlated with smoking, and overexpression of both the EGFR and ERα in NCLSC patients was associated with poor prognosis, establishing it as a useful prognostic factor in lung cancer [108]. Another report demonstrated that the total prevalence of ERβ overexpression was 45.8% (138/301) and female patients had the highest rate of detection (54.3 percent of 127 tumors versus 39.7% of 174 tumors in males). It signified that in patients with surgically resected stage II and III NSCLCs, immunohistochemistry expression of ERβ can be employed as a prognostic predictor for patients with NSCLC. These findings might lead to the advancement of hormone treatment for lung cancer patients [109]. In an expression study on ERs in human lungs, several positive correlations showed that ERα expression is more common in the lungs of women than men, whereas ERβ is found in the lungs of both men and women in almost similar proportions, and tumors contain a higher prevalence of both receptor types than non-tumors in women alone [110]. In NSCLC, ERβ expression increased and had an inverse relationship with lymph node metastasis, suggesting that ERβ negativity could be correlated with malignant progression of NSCLC [111]. ERβ expression was substantially greater in ADK tissues and cell lines than in SCCs, also suggesting that ERβ, but not ERα, is detected in lung tissues and has a physiological role in proper lung function. Furthermore, ERβ may also have a role in ADK growth and development [100]. In another study, lung cancer tissues were shown to exhibit greater ERβ expression compared to normal lung tissues [112,113,114], and a higher level of cytoplasmic ERβ expression/low PR was reported to be an independent prognostic factor for patient survival [115]. Elevated IL6/ERβ expression in lung cancer was seen to be correlated to less differentiation and increased metastasis, and the expression of IL6 was suggested to be an independent predictive factor for overall survival (OS). Furthermore, when stimulated by E2, ERβ regulates IL6 expression through the MAPK/ERK and phosphatidylinositol-3 kinase (PI3K)/AKT pathways, promoting malignant tumors [114]. In an in vitro study in A549 and H1793 cells evaluating treatment with an ER antagonist, fulvestrant (Ful) inhibited cell growth, involving downregulation of IL-6 that was induced in response to estrogen. According to the findings of this study, ERβ/IL6 may represent prospective therapeutic targets for prognosis and interventions in lung cancer [114]. Researchers have determined that patients with tumor stages I–II have higher ERβ mRNA expression and reduced ERβ protein expression as tumor grade increases [116]. Lung ADK tissues were demonstrated to have elevated ERβ protein expression as compared to adjacent non-cancerous tissues. The ERβ protein expression was shown to be associated with tumor enlargement, lymph node metastasis, clinical stage, and differentiation [117]. Reports on suppressing ERβ have been carried out to understand the effects of targeting ERβ on cancer cells, and an in vitro study showed that ERβ knockdown can suppress colony formation and cell invasion. In addition to that, an in vivo study demonstrated a reduced number of metastatic tumors in the lung of mice as a result of decreased expression of phosphorylated extracellular signal-regulated kinase (pERK), matrix metalloproteinase (MMP)-2, and MMP-9. These findings indicate that the ER could serve as an important player in lung ADK progression through mitogen-activated protein kinase kinase (MEK)/extracellular signal-regulated kinase (ERK) signaling, and suggested that the ER can provide a potential therapeutic target for lung ADK in the future [117]. A report elucidated that siRNA-mediated ERβ1 treatment may reduce the activity of pERK1/2 and pAkt in PC9-ER and HCC827-ER cells. This study indicated that ERβ1 upregulation due to resistance may stimulate ERK1/2 and Akt pathways [118]. Additionally, short hairpin RNA-mediated knockdown of ERβ stopped cell proliferation and caused apoptosis regardless of estrogen treatment, indicating that ERβ has a ligand-independent role in controlling the intrinsic apoptotic pathway [119]. In another study, treatment with estradiol-17β (E2β) showed increased pMAPK signaling and subsequently increased cell growth, and also secretion of VEGF. These effects exerted by E2β were reversed by the treatment with Faslodex, a pure antiestrogen. This report suggests that Faslodex, in vitro, is an efficient antitumor treatment, but when combined with human epidermal growth factor receptor (HER) receptor inhibitors, it has a stronger growth-suppressive impact [120]. Thus, combination therapy targeting ERs and growth factor receptor (GFR) signaling pathways might result in a more potent and long-lasting antitumor impact. E2 therapy was shown to be able to bring back ER mRNA expression and reduce ER hypermethylation in lung cancer A549 cells [121].

Another study involving estrogen-induced stimulation of pEGFR and phospho-p44/MAPK showed non-nuclear ER transactivation of the EGFR in NSCLC cells. In vitro, EGFR protein expression was suppressed in response to estrogen and upregulated in reaction to Ful, indicating that the EGFR pathway is triggered in NSCLC cells when estrogen is deficient. Targeting both the ER and the EGFR with an ER antagonist, Ful (“Faslodex”), and the selective EGFR tyrosine kinase inhibitor, gefitinib (“Iressa”), exhibited decreased proliferation of up to 90% and increased apoptosis [122]. The same study, using an in vivo xenograft model, found that combining Ful and gefitinib reduced tumor volume in SCID mice. These findings show that the ER and EGFR pathways interact in a functional manner in NSCLC [122]. Another report demonstrated that targeting both ERβ and toll-like receptor 4 (TLR4) might be a unique therapeutic strategy for advanced metastatic lung cancer. It was because estrogen increased NSCLC metastasis through the ERβ by upregulating TLR4 and activating the myeloid differentiation factor 88 (myd88)/nuclear factor kappa B (NF-κB)/MMP-2 signaling axis in vitro [123]. Targeting either ERα or ERβ is considered to be a potential approach because an in vitro study on H23 cells showed that knocking down either subtype exerts a significant reduction in NSCLC cell proliferation. The EGFR and ERs have a role in boosting early p42/p44 MAP kinase activation. With the purpose of inhibiting NSCLC development, Faslodex was administered in combination with erlotinib, which suppressed NSCLC xenograft development in vivo [124]. Clinical trials are now underway to investigate the application of antiestrogens alone and in combination with GFR antagonists. A study showed that the treatment with gefitinib combined with tamoxifen suppressed the growth and enhanced the apoptosis of A549 and H1650 ADK cell lines [125]. Based on studies cited, the ER is an established viable target in lung cancer as its expression is deregulated in various lung cancer cell lines and tissues. A plethora of research has been carried out on targeting ERs in lung cancer using different ER-specific antagonists, which has shown reliable results in inhibiting cancer cell growth and inducing apoptosis.

### 2.3. Peroxisome-Proliferator-Activated Receptors (PPARs)

The NR superfamily includes PPARs, which are ligand-dependent TFs. PPAR-regulated genes have a significant role in cell differentiation as well as metabolic functions such as glucose and lipid metabolism [126,127]. When PPARs are activated by a ligand, they heterodimerize and form complexes with coactivators such as p300, CBP, or SRC-1, which then attach to specific DNA regions called peroxisome proliferator response elements (PPREs) on the promoters of target genes, resulting in gene activation and repression [128,129,130,131,132]. In absence of a ligand, corepressors such as the nuclear receptor corepressor (NCoR), receptor-interacting protein 140 (RIP140), or silencing mediator of retinoic acid and thyroid hormone receptor (SMRT) form a complex with PPAR, thereby transcriptionally repressing target genes [133,134,135]. The PPAR family has three members: PPARα, PPARβ/δ, and PPARγ, which differ in their functions, expression, locations, and ligand specificities. PPARα expression is abundant in the liver, smooth muscle cells, enterocytes, intestinal mucosa, endothelial cells, heart, and immune cells such as monocytes/macrophages and lymphocytes [136,137]. It also has a role in fatty acid metabolism [138]. PPAR/β/δ is found in abundance in skeletal muscle, adipose tissue, the kidney, liver, gut, lungs, brain, and skin, where it regulates lipid metabolism. PPARγ, which regulates glucose and lipid metabolism, is expressed in white and brown adipose tissues, the large intestine, and the spleen, with the greatest levels of expression in adipocytes [131,139,140,141,142,143]. The nucleus houses PPARs, which form heterodimers with the retinoid X receptor (RXR) (Figure 3C). The N-terminal region of PPARs acts in transcriptional activation, while the DBD comprises two zinc-finger motifs that aid in heterodimer formation with RXR [144]. The interaction of CTD with proteins is aided by a hinge region in its structure. The CTD is also known as an LBD with a ligand-dependent AF-2 that is important in the formation of heterodimers with RXR and other cofactors [145,146].

Extensive research has been conducted on PPARγ as a target using various agonists and antagonists. Both human NSCLC and SCLC cells exhibited PPARγ mRNA and protein, and reported that ligands such as ciglitazone and 15-deoxy-δ-12,14-prostaglandin J2 (15d-PGJ2) suppressed the growth of cancer cells at particular doses or times, with ciglitazone being less effective than 15d-PGJ2. Treatment with ciglitazone and 15d-PGJ2 also caused apoptosis in lung cancer cells [147]. The mRNA expression levels of PPARγ were observed to be reduced in lung cancer tissues compared to the normal adjacent lung tissues. Patients with NSCLC who had low PPARγ mRNA expression had a substantially worse survival rate than the patients without low PPARγ mRNA levels. As a result, PPARγ mRNA levels might be used as a prognostic indicator in lung cancer, and the measurement of PPARγ mRNA levels might be useful as a marker for PPARγ agonist therapy of lung cancer [148]. Following this, the findings from a report show that primary tumors from NSCLC patients have higher PPARγ expression than normal adjacent tissue, and PPARγ expression was also found in numerous NSCLC cell lines. Troglitazone (Tro), a PPARγ ligand, increased PPARγ activity and inhibited cell proliferation by inducing reduction in G1 cyclins, D and E in a dose-dependent manner in ADK cell lines. PPAR ligands elicited persistent Erk1/2 activation, implying that it involves stimulation of a differentiation-inducing pathway. This study also supports the theory that PPARγ activation suppresses lung tumor growth and suggests that PPARγ ligands might be used as NSCLC therapeutics [149]. The inducible nitric oxide synthase (iNOS) and PPARγ have been linked to cancer development, and studies have shown higher and lower expression of the iNOS and PPARγ, respectively, in tumorous tissues than in non-tumorous tissues. Statistically, the levels of iNOS and PPARγ were revealed to be negatively associated. This report demonstrates that iNOS and PPARγ expression in NSCLC tissues vary. Because active PPARγ can decrease iNOS expression, and iNOS production is linked to inflammatory and environmental variables that increase lung cancer risk, this difference in the expression pattern could be attributed to NSCLC etiology [150]. In one study, human lung cancer cell lines were shown to express PPARγ but not PPARα. It also showed that thiazolidinedione (TZD) compounds (Tro and 15d-PGJ2) can inhibit the proliferation of lung cancer cells by inducing apoptosis. This shows that PPARγ may have an important role in lung cancer development, and that PPARγ agonists can be helpful therapeutic agents as a therapy for lung cancer [151].

A report showed positive PPARγ immunostaining in 61 out of the 147 cases (42%), whereas the rest were found to be negative [152]. In an in vivo study, PPARγ and PPARα expression was substantially greater in NNK-induced lung tumor tissues than in normal lung tissues. Treatment of mice with PIO dramatically resulted in the reduction in NNK-induced mouse lung tumors, indicating that stimulating PPARγ activity with its ligands and suppressing PPARα activity with inhibitors may help to prevent lung tumor growth. This might help in establishing tumor markers for lung cancer based on the level of endogenous PPAR ligands and the activities of PPARγ or PPARα [153]. In precancerous human bronchial epithelial cells (HBECs), PPARγ showed elevated expression, and increased expression of pro-inflammatory cyclooxygenase 2 (COX2) was reversed after PPARγ activation by TZD therapy. This therapy also suppressed tumor cell proliferation, clonogenecity, and migration [154]. In a study, the expression of PPARβ/δ was found in all NSCLCs investigated [155]. PPARβ/δ activation with ligand GW501516 increases cell proliferation, anchorage-independence of cell growth, and suppresses cell death in lung cancer cell lines by inducing Akt phosphorylation, upregulating of pyruvate dehydrogenase kinase 1 (PDK1), reducing phosphatase and tension homolog (PTEN), and enhancing expression of B-cell lymphoma-extra-large (Bcl-xL) and COX2. These results suggest that PPARβ/δ promotes cell proliferation and prevents apoptosis via the phosphatidylinositol-3 kinase (PI3K)/Akt1 and COX2 pathways. Finally, PPARβ/δ expression is abundantly found in many lung malignancies, and its stimulation promotes proliferation and survival responses in NSCLC [155]. Studies that used ligands such as GW0742 and GW501516 found out that they activate PPARβ/δ and induce an increase in a PPARβ/δ target gene, angiopoietin-like 4 (Angptl4), and had no effects on PTEN expression or Akt phosphorylation, thereby not affecting cell growth [156]. However in another study, treatment with GW501516, a PPARβ/δ agonist, enhanced prostaglandin E receptor 4 (EP4) expression and promoted NSCLC proliferation, demonstrating that PPARβ/δ activation may be a unique molecular strategy to control the development of human cancer [157].

PPARγ expression was studied in the two lung cancer cell lines. This study demonstrated that ligands such as PGJ2 and ciglitazone can stimulate PPARγ to suppress cell proliferation and cause death involving upregulation of caspase-3, Bcl-2-associated X (bax), and B-cell lymphoma 2 (Bcl-2), representing their role in these processes. PPARγ, which has a crucial role in the development and/or advancement of lung cancer, might be a promising new therapeutic target [158]. Fibronectin (Fn) is assumed to have a function in tumor cell invasion due to its expression in cancer cells [159,160]. In NSCLC, Fn expression was also elevated [161,162]. The adherence of lung cancer cells to Fn promotes tumorigenicity and imparts resistance to chemotherapy induced apoptosis [163]. The use of 15d-PGJ2, rosiglitazone (BRL49653), or Tro suppresses Fn gene expression in NSCLC by lowering the binding activities of the TFs cAMP response element (CRE) and specificity protein 1 (Sp1) via the PPAR pathways, and these inhibitory effects of ligands are abrogated by a PPARγ antagonist, GW-9662 [164]. Ciglitazone and sulindac sulfide, a COX inhibitor, target PPARγ expression and induce E-cadherin expression, and were associated with increased NSCLC differentiation. Sulindac sulfide greatly slowed tumor growth in nude mice bearing the A549 NSCLC line. As a result, nonsteroidal anti-inflammatory drugs (NSAIDs) such as sulindac sulfide are potent inhibitors of NSCLC cell transformation [165].

Induction of PPARγ expression in H2122 ADK cells (H2122- PPARγ) inhibited anchorage-independent growth. Implantation of these cells into the lungs of nude rats was shown to exert inhibitory effects on tumor growth and metastasis, suggesting the inhibitory effects of PPARγ on lung tumorigenesis involve selective inhibition of invasive metastasis [166]. The growth arrest- and DNA-damage-inducible gene 153 (GADD153) was shown to be a potential factor involved in NSCLC cell growth suppression and apoptosis through TZD-induced PPARγ activation [167]. The effects of thalidomide on NSCLC cell proliferation inhibited the growth of LCC cells, and this compound enhanced PPARγ protein, and PPRE at the molecular level. Furthermore, thalidomide treatment of LCC cells reduced NF-κB activity and angiogenic protein production and enhanced apoptosis [168]. In a xenograft model, researchers discovered that thalidomide reduced tumor development by 64%, and tumors in thalidomide-treated mice had higher expression of PPARγ than tumors in control mice. This research demonstrates the anticancer effects of thalidomide against LCC tumors and proposes a model in which thalidomide acts on LCC cells by inducing PPARγ and subsequent downstream signaling [168]. A study elucidated that Tro can hamper the growth of lung cancer cells via inducing apoptosis and, at least in part, inhibit cell proliferation in a PPARγ-relevant manner. Tro led to the activation of ERK and p38, the reduction in stress-activated protein kinase (SAPK)/c-Jun NH2-terminal kinase (JNK) activity, and the reduction of Bcl-w and Bcl-2 [169]. The reduced growth by PPARγ ligands such as GW1929, PGJ2, ciglitazone, Tro, and rosiglitazone was related to a substantial reduction in EP2 mRNA and protein levels in human NSCLC cell lines (H1838 and H2106). GW9662, a selective PPARγ antagonist, abolished the inhibitory effects of BRL49653 and ciglitazone, but not PGJ2. This study reveals that PPARγ ligands suppress lung cancer cell proliferation by reducing EP2 receptor expression via ERK signaling and PPARγ-dependent and -independent pathways [170]. Another report showed that PGJ2 and ciglitazone impede lung cancer cell proliferation and induce apoptosis by activating the cyclin-dependent kinase (CDK) inhibitor p21 and lowering CD1 gene expression. Increased Sp1 and the nuclear factor for IL-6 expression (NF-IL6) CCAAT/enhancer binding protein (C/EBP)-dependent activation may lead to PPARγ ligand-induced elevation of p21 gene expression, unveiling a process for p21 gene regulation in lung carcinoma that could be a potential lung cancer therapy [171]. Tro significantly upregulated PPARγ and caused apoptosis in NCI-H23 lung cancer cells via a mitochondrial mechanism that was PPARγ- and ERK1/2-dependent. However, PPARγ siRNA abrogated these effects of Tro [172]. 

Many studies have shown the potential role of PPARγ ligands as therapeutic agents against NSCLC [173]. Researchers have shown that combined expression of Wnt family member 7A (Wnt 7a) and Frizzled 9 (Fzd 9) in NSCLC cell lines inhibits malignant growth. Expression of Wnt 7a and Fzd 9 led to ERK5 activation that in turn caused PPARγ activation in NSCLC. SR 202, a PPARγ inhibitor, inhibited the increase in PPARγ activity and restored the anchorage-independence property of cells. These findings imply that ERK5-dependent PPARγ activation is a significant effector mechanism driving the anticancerous actions of Wnt 7a and Fzd 9 in NSCLC [174]. According to a study, in vitro administration of 15d-PGJ2 and docetaxel had a synergistic interaction against A549 and H460 cancer cell lines. In a xenograft athymic nu/nu mice model, 15d-PGJ2 in combination with docetaxel significantly lowered the tumor compared to other groups. A considerable increase in apoptosis was seen in 15d-PGJ2 as well as docetaxel-treated cells, which was linked to suppression of Bcl2 and CD1 expression, as well as upregulation of caspase and p53 pathway. Furthermore, GW9662 did not reverse the increased level of caspase 3, and inhibition of CD1 was caused by 15d-PGJ2 as well as docetaxel, suggesting a putative PPARγ-independent mechanism. Hence, 15d-PGJ2 increased the antitumor activity of docetaxel through apoptosis induction mediated by PPARγ-dependent and -independent processes [175]. A novel PPARγ agonist, 1-(trans-methylimino-N-oxy)-6-(2-morpholinoethoxy)-3-phenyl-(1H-indene-2-carboxylic acid ethyl ester (KR-62980), and rosiglitazone, decreased the viability of NSCLC cells by the induction of PPARγ while having a minimal effect on SCLC cells [176]. According to the findings, PPARγ activation causes apoptotic cell death in NSCLC mostly by the production of reactive oxygen species (ROSs) through the induction of peroxidase (POX), a redox enzyme found in mitochondria [176]. Another study reported that TZDs, Tro, and ciglitazone increased vascular endothelial growth factor (VEGF) and neuropilin-1 expression and inhibited cell proliferation. Researchers hypothesize the existence of a cell growth arrest mechanism involving the interaction of TZD-induced VEGF and neuropilin-1 in NSCLC [177].

The US Food and Drug Administration (FDA) has approved the use PIO for treating type 2 diabetes. It belongs to the TZD family of antidiabetic medicines. In the lung ADK model, PIO significantly reduced tumor load (average tumor volume per lung) by 64% in p53 (wt/wt) mice and 50% in p53 (wt/Ala135Val) mice, demonstrating that in both the ADK and SCC mouse model systems, PIO suppressed tumor development [178]. After treatment with PIO, the proliferative and invasive properties of NSCLC cells were inhibited, and apoptosis was induced, involving downregulation of MAPK, Myc, and Ras genes. By suppressing cell growth and invasion via blockage of MAPK and transforming growth factorβ (TGFβ)/SMADs signaling respectively, one study demonstrated PPARγ agonists as a promising therapy option for NSCLC [179]. In another study, ciglitazone showed significantly suppressed A549 proliferation and tumor growth in nude mice. In the ciglitazone-treated group, PPARγ expression was significantly increased, involving decreased expression of CD1 and enhanced p21 levels [180]. Another study found that a novel PPARγ agonist, CB13 (1-benzyl-5-(4-methylphenyl) pyrido [2,3-d]pyrimidine-2,4(1H,3H)-dione), inhibits cell growth by reducing cell viability, elevating lactate dehydrogenase (LDH) release, and enhancing caspase-3 and caspase-9 activity [181]. In human NSCLC cells, the purine-based PPAR ligand CB11 (8-(2-aminophenyl)-3-butyl-1,6,7-trimethyl-1H-imidazo [2,1-f]purine-2,4(3H,8H)-dione mediates cell death, ROS production, mitochondrial membrane potential disintegration, and cell cycle arrest [182]. CB11 significantly reduced tumor volume in a xenograft experiment when compared to a control group. These effects of CB11 support its potential as an anticancer agent, and it can be employed to control radio-resistance in NSCLC patients who have been exposed to radiation [182]. Treatment with Les-2194 and Les-3377 in the SCC-15 and CACO-2 cell lines increased ROS generation, but had no effects on caspase-3 activity or metabolic activity. Despite this, the level of Ki67 dropped considerably. Les-3640, on the other hand, was able to cause enhanced production of ROS in BJ, SCC-15, and CACO-2 while having no effect on metabolic activity. However, all of the cell lines tested showed increased caspase-3 activity [183]. An investigation of NSCLC cells reported a novel PPAR ligand candidate, PPZ023 (1-(2-(ethylthio) benzyl)-4-(2-methoxyphenyl) piperazine), that decreases cell viability while increasing LDH cytotoxicity and caspase-3 activity. By producing ROS and inducing mitochondrial cytochrome c release, PPZ023 can induce apoptosis [184]. Ciglitazone and the RXR ligand SR11237 decreased the proliferation of Calu-6 lung cancer cells in a cooperative manner. When ciglitazone was combined with SR11237, it significantly increased RARβ expression in lung cancer cell lines, which was diminished by a PPARγ-selective antagonist, bisphenol A diglycidyl ether. These findings suggest that PPARγ action is regulated via a unique RARβ-mediated signaling pathway, which could provide a biological foundation for developing new therapeutics using RXR and PPARγ ligands in potentiating antitumor responses [185].

PPARδ is known to play a functional role in glucose and lipid homeostasis. Carbaprostacyclin, a prostaglandin (PG)I2 agonist for IP and PPARδ, and L-165041, a PPARδ agonist, were used and shown to exert negative growth control of A549 using PPARδ as a critical molecule of PGI2 signaling. This suggests that PPARδ activation in the presence of suppressed PG synthesis is critical for the regulation of growth of lung cancer cells [186]. Another study showed that the treatment of a lung cancer cell line, A549, with SR13904, a PPARδ antagonist, leads to significantly lower levels of a variety of cell cycle proteins. PPARδ is known to upregulate several of these cell-cycle-related genes, such as cyclin A and D, and CDK 2 and 4. The anticancer properties of SR13904 show that blocking PPARδ-mediated transactivation could reduce carcinogenesis, and that blockage of PPARδ could be a promising cancer therapy or prevention method [187]. Results from the extensive research conducted on PPARs over the years have elucidated that PPAR expression is altered in lung cancer, especifically PPARγ that has been targeted using varieties of agonists and antagonists in order to treat different lung cancer cell lines in vitro and models in vivo. Various available reports suggest that these PPAR ligands have been able to reduce tumor load and growth and have antiangiogenic and antiproliferative properties, owing to which PPAR could be proven as a potential target to combat lung cancer using different ligands and small molecules that can be designed with an optimized procedure.

### 2.4. Retinoic Acid Receptors (RARs)

RARs are ligand-dependent TFs that work in partnerships with RXRs to form heterodimers. Vitamin A derivatives activate RARs that are important regulators of genes involved in cellular proliferation, differentiation, and death [188,189,190]. RARs have been linked to diverse biological tasks, including development, reproduction, immunity, organogenesis, and homeostasis [191,192,193,194]. Retinoids, such as vitamin A metabolites and synthetic ligands, are ligands that activate RARs [195]. RARs are divided into three subtypes: α (NR1B1), β (NR1B2), and γ (NR1B3), each of which is encoded by a different gene (Figure 3D). RARα is expressed all across the body; however, RAR β and RAR γ exhibit tissue-specific expression patterns [193]. The RAR structure is made up of three functional domains: an unstructured and non-conserved A/B NTD, a C-region-containing DBD, and a hinge in the D-region that connects DBD to a C-terminal LBD [196,197]. It has two zinc-finger domains, a ligand-binding site, a dimerization domain, and a hydrophobic site for coregulatory binding found in the DBD. Upon binding to a ligand, RARs bind to specific DNA sequences in the promoter region of target genes, causing conformational changes in the LBD that aid in transcriptional control of coregulators [195,196,197]. RARs also work along with RXR to form heterodimers that bind to RA response elements (RAREs) in gene promoters [198].

Various studies have shown the involvement of RARs in many malignancies, including lung cancer. RARβ is the most well-known of the three RARs for its role as a tumor suppressor in epithelial cells [199,200,201]. The exogenously expressed RARβ gene has been shown to cause apoptosis and growth arrest involving RARα in both an RA-dependent and -independent manner [198,202]. Many activators assemble at the RARE on the RARβ promoter as RARα bound to RA binds to the RARE, resulting in upregulation of the RARβ gene. Previous research has shown that RARβ expression is reduced in lung cancer in vitro and in vivo, implying that RARβ could have tumor-suppressive effects in lung carcinogenesis [203,204,205,206]. Reports have shown the deregulated expression of RARs in lung cancer tissues and various cell lines. One such study determined the expression of RARs in normal and malignant tissues from 76 patients with NSCLC and showed that tumor cells have overexpressed RXRα and RARα and reduced RXRβ, RARβ, and RARγ [207]. Lung cancer cell line, EBC-1, was shown to have upregulated RARα, and it was suggested that the growth inhibitory effects of vitamin A are linked to RARα mRNA expression [208]. RARα and RARγ expression was detected, but RARβ was not found to be expressed in Calu-1 cells [209]. In vitro analysis from the same study showed that using all-trans-retinoic acid (ATRA) can inhibit malignant growth of lung cancer cell lines [209]. Findings of another study demonstrated that RARα and RARγ are expressed in the rat tracheobronchial epithelial cell line SPOC-1 and showed ATRA to be a strong stimulator of tissue transglutaminase (TGase II) and apoptosis in these cells. An RAR-selective retinoid, SRI-6751-84, and the RARα-selective retinoid Ro40-6055 can enhance TGase II expression, which is completely abrogated by the RARα-antagonist Ro41-5253, implying that an RARα-dependent signaling pathway is involved in the activation of TGase II production and apoptosis in SPOC-1 cells [210]. In a study, RARα and RARβ were shown to be downregulated in malignant lung tissues compared to normal tissues, and the levels of RARβ mRNA and alcohol dehydrogenase 3 mRNA were shown to be significantly correlated. These findings suggest a link between lung carcinogenesis and the loss of RARα, RARβ, and alcohol dehydrogenase 3 [211]. Reduced expression of subtypes of RAR and RXR is a common occurrence in NSCLC, and assessment of RAR and RXR mRNA expression in tumor tissues is a potential predictive and surrogate biomarker for NSCLC chemoprevention trials, according to a published report [212]. Additionally, depletion of RARs has been shown to be linked to a worse prognosis, and that these receptors could be a molecular target for NSCLC [213]. The mRNA expression of genes of RXRγ, RARα, and RXRα was shown to be considerably lower in tumor specimens, revealing that deregulation of these genes follows the aberrant transformation of lung tissue cells [214]. The aberrations of the RARβ network are frequent in human lung cancer cell lines [215]. BEAS-2B-R1 cells had hypermethylation in RARβ P2 promoter and showed lower RARβ1 expression, and treatment with azacitidine restored the expression of RARβ2 and PTGFβ. These findings suggest that in lung carcinogenesis, restoring RARβ1 expression may overcome retinoid resistance [216]. In contrast to all these findings, a report demonstrated the higher RARβ expression in cancer tissues, with a localized and uneven distribution in normal-appearing surrounding bronchial epithelium and inconsistencies with tumor tissues. The data suggest that RARβ expression could be used as a biomarker for chemoprevention/early diagnosis or the prognosis of NSCLC and warrants further investigation [217]. Lung cancer cell lines and bronchial biopsy specimens have an aberrant expression of RARβ [205,218]. Treatment of bronchial biopsy specimens and lung cancer cell lines with 13-cis-RA (13cRA) showed an increase in RARβ expression [218,219].

RARβ expression is often downregulated in the bronchial epithelium of smokers, but is increased after 13cRA therapy [220]. From the findings of a study, it was implied that RARα, RXRα, and RXRγ expression is unaffected in NSCLC. However, RARβ and perhaps also RARγ and RXRβ were repressed in a high proportion of lung cancer patients, suggesting the association of lung carcinogenesis with the loss of expression of one or more receptors [206]. A report showed that RARβ is increased in expression in lung cancer cell lines, and CD437, which binds selectively to nuclear RARγ, can hamper cell growth and cause apoptosis in ATRA-resistant NSCLC cells [221]. A study reported an in vitro increased expression of RARα and RARγ and reduced RARβ in lung carcinoma cell lines [222]. In vivo NNK-induced animal models have been shown to have reduction in tumor multiplicity and an increase in RARs levels [223]. In a study, retinoic acid (RA) was found to exert the growth inhibitory effects involving the upregulation of p27 (Kip1) and the downregulation of CDK3 and p21 (CIP1/Waf1). Moreover, the growth arrest was linked with enhanced RARβ and decreased c-Myc expression [224]. A study revealed that RA can elicit gastrin-releasing peptide (GRP), a growth factor that can act as a tumor promoter, by activating intact retinoid signaling, implying that retinoids interestingly may enhance rather than decreasing the risk of lung cancer in some people [225]. Treatment of NSCLC cell lines with a novel retinoid 6-[3-(1-adamantyl)-4-hydroxyphenyl]-2-naphthalene carboxylic acid (CD437) was reported to stimulate apoptosis via a mechanism that is independent of NRs involving upregulation of c-Myc and its downstream targets ornithine decarboxylase (ODC) and cdc25A, showing the c-Myc gene to be crucial in triggering CD437-induced apoptosis [226]. The mechanism of apoptosis induced by CD437 involves induction of p53 and its target genes, p21, Bax, and Killer/DR5, in lung cancer cells. CD437 can potentially cause apoptosis through a mechanism that is not dependent on p53 [227]. An in vitro study on lung ADK cells revealed that RA- or 4-HPR-treated cells show halted growth by increasing RARβ2 [228]. The studies show that RA promotes the synthesis of EGFR in human lung cancer cells, which is reflected by their rising tumorigenicity phenotype [229,230]. A novel retinoid-regulated gene, tazarotene-induced gene 3 (TIG3), also known as retinoic acid receptor responder 3 (RARRES3, accession number AF 060228), has tumor-suppressive effects [231]. The expression of TIG3 was reported to be increased in response to the treatment with ATRA, and this increase is linked to the reduction in anchorage-independent growth [232]. β-Cryptoxanthin (3-hydroxy-β-carotene), a pro-vitamin A carotenoid, was shown to reduce the proliferation of A549 and BEAS-2B cells by involving reduction in CD1 and cyclin E and an increase in cell cycle inhibitor p21. In BEAS-2B cells, this compound increased RARβ mRNA levels, although this impact was less significant in A549 cells, suggesting that this compound could be a promising chemopreventive drug for lung cancer [233]. ATRA therapy enhanced the protein and mRNA levels of VEGF-C and VEGF-D, and their receptor VEGFR3 gene in a dose-dependent manner, which was correlated with higher levels of RARα expression and lower levels of another ATRA receptor, PPAR β/δ [234]. The findings from a study show that N-(4-hydroxyphenyl)retinamide (4HPR) is more effective than ATRA at inducing apoptosis in NSCLC cells, implying that more clinical trials using 4HPR for NSCLC prevention and treatment are needed [235]. Through extensive research, RARs have been shown to be disturbed in expression. RARβ is one of the RARs which has tumor-suppressive properties and is found to be underexpressed, with RARβ hypermethylation being the major reason in lung cancer subjects. Several agonists and antagonists have been demonstrated to exert antitumor properties by targeting RARs, which in turn may lead to growth inhibition and apoptosis in cancer cells. There are extensive studies available targeting RARs in lung cancer, and there are many possibilities of finding novel and reliable therapeutics that could be helpful as safe and efficacious approaches against lung cancer.

### 2.5. Retinoic X Receptors (RXRs)

RXRs are TFs that function as homodimers and heterodimers in combination with other NR families, such PPARs, RARs, FXRs, LXRs, TRs, CAR, NURR1, and vitamin D3 receptors (VDRs) [236,237,238,239]. RXRs are divided into three subtypes: RXRα (NR2B1), RXRβ (NR2B2), and RXRγ (NR2B3), each of which is encoded by a different gene. Permissive and non-permissive heterodimers are formed by RXRs. RXR ligands alone activate permissive heterodimers [240]. Non-permissive cells, on the other hand, are not transcriptionally activated by the RXR ligand alone but require the assistance of partner ligands [241,242,243]. RXRα is found in the liver, muscle, kidney, lung, epidermis, and intestine, whereas RXRβ is widely expressed. RXRγ1 and RXRγ2 are two RXRγ isoforms found in separate tissues, with RXRγ1 found in the brain and muscle and RXRγ2 found in the cardiac and skeletal muscles [197]. RXRs and RARs have a considerable difference in that RXRs are activated by agonist 9cRA, while RARs bind to RA and 9cRA. The structural domains of RXRs are divided into six groups (Figure 3E). The ligand-dependent/independent transcriptional AF-1 is found in the N-terminal region of the A/B region [197]. The DBD plays a role in recognizing specific sequences in the target gene and is located in the C region. The D domain serves as a hinge, connecting the C domain to the LBD in the E region. A partner dimerization site, a ligand-binding pocket (LBP), a site for coregulator binding, and a ligand-dependent AF-2 make up the LBD [197]. Ligand binding to an LBP stimulates a conformational change in the protein, which activates RXRs and recruits coregulators regulating target gene transcription [244]. After activation, RXR homodimers bind to RXR response elements (RXREs) in target genes, while RXR heterodimers bind to target genes through specific DNA sequences called HREs. In tumor specimens from the lung, the mRNA levels of RXR genes RXRα and RXRγ were shown to be considerably lower [214]. Another study reported the overexpression of RXRα in cancer cells, and RXRβ expression, on the other hand, reduced in 18% of tumor samples. These results suggest that RXR expression changes might have a role in the development of lung cancer [207].

The widespread coregulation of expression of all retinoid subclasses shows that the retinoid pathway is deregulated in lung malignancy. The mRNA expression levels of RAR and RXR in tumor tissues have been quantified as a possible marker for prognosis and a surrogate biomarker for chemoprevention trials in NSCLC [212]. In comparison to normal lung tissues, the median mRNA expression levels of all three RXR subtypes are typically lower in tumor samples. Those with low RXRβ expression levels have a statistically significant lower overall survival. Hence, mRNA expression of all three RXR subtypes is often suppressed, and reduced RXRβ expression in NSCLC patients could be a biomarker for more aggressive disease [245]. Mice given 9cRA supplementation exhibited significantly decreased tumor multiplicity and a trend toward lower tumor incidence, implying that 9cRA may protect against lung cancer, with this effect mediated in part by 9cRA activation of RARβ but not COX2 transcription suppression [246]. Treatment with LGD1069, an RXR agonist, decreased growth and CD31 expression in A549 xenograft models when compared to the control, and reduced density of the capillary network induced by tumor cells. Moreover, LGD1069 could be used to treat NSCLC by inhibiting tumor-induced angiogenesis [247]. MSU42011, a new RXR agonist, was shown to significantly decrease the tumor burden in a vinyl carbamate-induced A/J mouse model, suggesting that MSU42011 could be useful in modulating the tumor microenvironment in lung cancer [248]. Bexarotene, a RXR agonist suppressed proliferation and migration while increasing apoptosis in NSCLC cells. Furthermore, under the administration of bexarotene, increased slc10a2 in NSCLC cells can decrease proliferation and migration while promoting apoptosis. The enhanced expression of slc10a2 triggered the expression of PPARγ, which ultimately caused an increase in PTEN expression and decrease in mTOR expression, implying that bexarotene decreases lung cancer cell survival through the slc10a2/PPARγ/PTEN/mTOR signaling pathway [249]. RXRs are observed to be suppressed in lung cancer, and different agonists have reduced lung cancer development and progression by inhibiting cancer cell growth and metastasis, promoting cell death, and influencing the tumor microenvironment.

### 2.6. Vitamin D Receptors (VDRs)

VDR is a ligand-dependent TF that belongs to the NR superfamily [237]. VDR controls a variety of biological activities, including cell proliferation, differentiation, development, homeostasis, and a diverse physiological function. The active form of vitamin D, 1,25-dihydroxyvitamin D3 (1,25(OH)2D3), has a key role in calcium and phosphate metabolism. It exhibits immunosuppressive effects and is involved in cell differentiation induction [250,251]. VDR is made up of an A/B region with a variable NTD, a C region with a conserved DBD, a D region with a hinge, and an E/F region with an LBD [237,252]. A brief A/B region with no AF-1 domain exists in the VDR [253]. Despite this, the LBD possesses a dimerization interface with a ligand-dependent transcriptional AF-2. Then, AF-2 undergoes a conformational shift in response to ligand binding that aids in the recruitment of coactivators from the p160 or DRIP/TRAP families (Figure 3F) [254]. VDR heterodimerized with RXR, binds to a specific DR3 response sequence in the promoter of target genes, resulting in transcriptional activation or inhibition [251].

The VDR is expressed at high levels in tissues such as the gut, bone, and kidney [255]. However, research on pleiotropic effects of vitamin D has found that the VDR is expressed in a variety of tissues, including malignant tumors [256]. A study showed that the VDR is often found on the mRNA level in lung cancer cell lines. The VDR appeared to be notably expressed in SCC and ADK, whereas receptor expression was found in just 1/4 of the cases evaluated in LCC and SCLC, which may be thought to be less differentiated [257]. In a study, calcitriol, a major biologically active metabolite of vitamin D, suppressed the proliferation of one of the cell lines studied, EBC-1, in a dose-dependent manner. VDR mRNAs were shown to be expressed at significant levels in EBC-1 cells, suggesting that growth inhibitory actions of the vitamins are linked to mRNA expression for VDRs [208]. Significantly higher levels of the VDR were observed in lung cancer tissues compared to normal lung tissues taken from patients [258]. In lung cancer patients with early-stage NSCLC, circulating 25-hydroxyvitamin D (25(OH)D) levels, as well as high vitamin D intake at the time of surgery, were linked to enhanced survival [259,260]. In the same cohort, VDR gene polymorphism was linked to better survival in early-stage NSCLC patients [261]. In another report, a higher level of nuclear VDR expression was linked to a better overall survival rate. The results show that the presence of a nuclear VDR may be a prognostic factor in NSCLC [262]. VDR expression was found in 64% (9 of 14) of ADKs and 67% (10 of 15) of SCCs. In LCC, VDR expression was generally not found (25%) [75]. Another report showed that calcitriol could be used as a chemo-preventive drug to prevent the development of lung cancer [263]. In comparison to controls, vitamin D decreased the expression of histidine-rich calcium-binding protein (HRC), H460 cell migration and proliferation, and increased cell death. In lung cancer, HRC and VDR expression were dramatically increased and downregulated, respectively [264]. A report showed that 1alpha,25-D(3) and 22-oxa-1alpha,25-D(3) inhibited the expression of MMP-2, MMP-9, VEGF, and parathyroid-hormone-related protein in LLC-GFP cells. These results suggest that inhibiting metastasis and angiogenesis-inducing activities in cancer cells is a primary mechanism underlying the anticancer effects of 22-oxa-1alpha,25-D(3), suggesting this compound could be useful for controlling metastasis in lung carcinoma [265]. From the findings of various reports, it can be demonstrated that VDR expression is deregulated in lung cancer cells and tissues, and using specific agonists for VDR may help to overcome lung cancer by inhibiting tumor cell growth and progression.

### 2.7. Liver X Receptors (LXRs)

LXRα (NR1H3) and LXRβ (NR1H2) are two isotypes of ligand-activated NRs. LXRα expression is found in metabolically active tissues such as the liver, fat, gut, kidney, skin, and macrophages, while LXRβ is found in all types of tissues [266]. LXRs control many genes involved in cholesterol homeostasis, including cholesterol absorption, storage, catabolism, and transport [266,267,268]. They also have a role in the regulation of cell growth in a variety of cell types [54,269,270].

LXR activation has antiproliferative effects because of the breakdown of growth signaling pathways and the stimulation of proapoptotic signals, according to pharmacological investigations on numerous cancer models, including prostate, colon, mammary, and skin cancer [54]. LXRs have been shown to diminish the cell cycle regulators, such as the S-phase Kinase-associated protein (SPK2) in cancer cell lines [271], while also inducing the expression of cell cycle inhibitors, such as p21 and p27 (CDK inhibitors) in prostate and ovarian cancer cells, with corresponding lowering levels in phospho-retinoblastoma (pRb) protein [272,273]. Furthermore, LXR activation slowed the progress of androgen-dependent cancers to androgen independent in mice models [273,274].

LXR agonists have been demonstrated in studies to suppress the growth of a range of cancer cells, including prostate, breast, ovarian, and colorectal cancer [54,270]. Cholesterol compounds, such as oxysterols and 24(S), 25-epoxycholesterol, as well as synthetic agonists such as GW3965 and T0901317, activate both LXRs [275]. In combination with RXRs, LXRs form a heterodimer that may be activated by RXR ligands, such as 9cRA [276]. With a corepressor complex, LXR/RXR heterodimers bind precisely to DNA sequences called LXR response elements (LXREs) in target genes, suppressing gene transcription. When a ligand binds, a conformational shift takes place in the heterodimer, allowing corepressor complexes to be released and coactivator complexes to be recruited, which causes genes to be transcribed. LXRs are a possible target for cancer therapy; agonists of LXRs will have an influence on the tumor microenvironment as well as the receptor state [67]. According to some findings, LXR agonist T0901317 was shown to render natural EGFR-TKI-resistant A549 human lung cancer cells in response to EGFR-TKI treatment which is LXRβ-dependent [67]. On the other hand, T0901317 has no effect on the H1650 cell line, which is naturally resistant to EGFR-TKI [67]. Another report confirmed that the combination of the LXR agonist T0901317 and gefitinib can suppress lung cancer migration and invasion in vivo in BALB/c nude mice and in vitro, and that this effect is likely due to inhibition of ERK/MAPK signaling [277]. In a study, T0901317 suppressed the invasion and migration of A549 cells, and found out that activating LXRβ, a subtype of LXR, can reduce MMP-9 expression. This study suggested that the activation of LXRs by T0901317 suppresses the invasion and metastasis of NSCLC through inhibiting the NF-κB/MMP-9 signaling pathway [278]. The LXR agonists, GW3965 and RGX-104, rendered NSCLC radiosensitivity in a homograft mouse model [279]. GW3965 has previously been found to decrease NF-κB transcriptional activity [280]. The findings of a report have supported that GW3965 could be an effective treatment for gefitinib resistance [281]. Myeloid-derived suppressor cells (MDSCs) are a heterogeneous population of immature granulocytic and monocytic cells that are important components of the tumor microenvironment (TME). They have been shown to stimulate tumor growth [282], metastasis [283], angiogenesis [284], therapeutic resistance [285], and have significant immunosuppressive action [286]. MDSC abundance in the TME was considerably reduced after LXR activation [279]. In vitro, RGX-104 treatment significantly increased MDSC apoptosis. Ultimately, by reducing MDSCs, which sensitizes NSCLC to radiotherapy, an LXR agonist can partially reverse the immunosuppressive effects of radiotherapy [279]. In vitro, a combination of an LXR agonist and gefitinib was shown to decrease Akt-NF-κB activation and prevented the production of apoptosis-related proteins. LXR ligands, on the other hand, had no effect on gefitinib-resistant lung cancer cells when used alone. In conclusion, the study found support for treating acquired tyrosine kinase inhibitor (TKI) resistance in NSCLC with a combination of treatments [287]. In an in vitro study, LXRα knockdown using siRNA (si-LXRα) dramatically increased the growth of HCC827-GR and PC9-GR cells. The combination of efatutazone, a highly selective oral PPARγ agonist, and the LXRα agonist T0901317, had a synergistic therapeutic impact on lung ADK cell growth and protein expression of PPARγ, LXR A, and ABCA1. The results show that efatutazone suppresses cell proliferation via the PPARγ/LXRα/ABCA1 pathway, and that when paired with T0901317, it has a synergistic therapeutic effect [288]. A variety of agonists have been reported to activate LXRs and lead to decreased cancer cell growth and induced cell death in vitro and in vivo in xenograft models. They can also affect the tumor microenvironment and alter tumor phenotypes, as shown by available studies.

### 2.8. Estrogen-Related Receptors (ERRs)

ERRs or NR3B is one of the groups of the ER-like subfamily (NR3). The three members of the ERR family are: ERRα, ERRβ, and ERRγ. ERRα is expressed in tissues such as the heart, kidney, intestinal tract, skeletal muscle, and brown adipose tissue, whereas ERRβ and ERRγ are expressed in the heart and kidney [31]. Glycolysis, oxidative phosphorylation, and the tricarboxylic acid (TCA) cycle are among the cellular metabolic processes regulated by ERRs [289]. ERRα regulates OPN (osteopontin) and WNT11 (wingless-type MMTV integration site family, member 11) to regulate cell growth, migration, and invasion [289]. ERR is made up of six separate domains: A, B, C, D, E, and F (Figure 3G). The NTD is found in the A/B region, which contains the ligand-independent transcriptional AF-1, a hinge in the D region, an E region, also known as the LBD, that contains the ligand-dependent transcriptional AF-2, and an F domain in the C-terminal region. The expression of target genes is controlled by AF-2 in conjunction with AF-1 [290,291].

ERRα has been shown in a number of studies to regulate the carcinogenesis and progression of cancers such as breast [292] and ovarian cancer [293]. The presence of ERRα in a number of human malignancies, including breast, ovarian, and colon tumors, is associated with poor prognosis [294,295,296]. In the case of NSCLC, limited evidence suggests that inhibiting ERRα with its inverse agonist XCT790 can stop cell proliferation by inducing ROS [178]. Increased levels of ERRα can cause A549 cells to proliferate and migrate [297]. The findings of another study reveal that ERRα was considerably enhanced in NSCLC cell lines and in clinical tissues, but not ERRβ or ERRγ [298]. Upregulation of ERRα can cause NSCLC cells to proliferate, migrate, and invade involving upregulation of IL-6, and when IL-6 was silenced, these effects were reduced [297,298]. Moreover, it was discovered that inhibiting NF-κB, but not ERK1/2 or PI3K/Akt, prevented IL-6 induced by ERRα. Overall, findings reveal that NF-κB/IL-6 activation is implicated in ERRα-induced NSCLC cell migration and invasion. It was proposed that ERRα could be targeted as a therapy for NSCLC [298]. From the various reports, it has been shown that ERRα, but not ERRβ or ERRγ, is upregulated in NSCLC, which can induce proliferation and migration of cancer cells, and inhibiting ERRα may be a promising strategy for lung cancer therapy.

### 2.9. Farnesoid X Receptor (FXR)

FXR is a ligand-dependent TF that is encoded by the NR1H4 gene. There are two FXR subtypes: FXRα (NR1H4) and FXRβ (NR1H5). FXRβ is found as a pseudogene in humans and FXRα in tissues such as the liver and small intestine and also in the kidney, adrenal glands, lung tissues, and blood vessels (Figure 3H). It regulates transcription of various genes responsible for bile acid synthesis, thereby maintaining bile acid homeostasis. FXR functions as a regulator of inflammation and immune response in immune diseases [299,300]. It regulates various genes by binding to a specific region FXR response element (FXRE) in the promoter of target genes [301]. FXR has a ligand-independent transcriptional AF-1, a core DBD, a hinge region, a C-terminal LBD, and a ligand-dependent AF-2 [302].

Besides the normal role, FXR has been shown to directly influence the growth of cancer cells as an oncogene, and can activate various other oncogenes, such as CD1, which was seen in NSCLC [303]. FXR may have a significant role in carcinogenesis as either an oncogene or a tumor suppressor gene. In mice, FXR deficiency has been shown to activate the Wnt/β-catenin pathway in the liver, leading to the development of liver tumors [304,305]. In other investigations, FXR−/− mice were found to have enhanced colon cell proliferation and small-intestine ADK development, showing human colorectal cancer progression is inversely associated with FXR expression [306,307]. FXR is dramatically raised in NSCLC, according to one study, and it predicts worse clinical outcomes in NSCLC patients [308]. FXR suppression in NSCLC cells decreased cell proliferation in vitro, stopped xenograft growth, and slowed the G1/S transition of the cell cycle, whereas overexpression of FXR increased cell proliferation in NSCLC cells. This study suggested that FXR may have an oncogenic role in the development of NSCLC, thereby giving molecular insights which could be used for prognostic and therapeutic purposes [308]. FXR has an important role in lung cancer development and progression. There is a need for more research on FXR in lung cancer to explore the possibilities of using it as a reliable therapeutic target in lung cancer, and also to seek the use of different FXR-specific agonists and antagonists in order to target FXR for the prevention of lung cancer.

### 2.10. Pregnane X Receptor (PXR) or NR1I2

PXRs are members of the NR family which function in a ligand-dependent manner and regulate various genes. The ligands needed for PXR activation include several endobiotics and xenobiotics [309]. It regulates the transcription of genes through binding to RXRα as a heterodimer (Figure 3I). Its expression is seen in many tissues such as the liver, small intestine, and colons in human. PXRs have an important role in the regeneration of the liver and regulating proliferation of both cancer and non-cancer cells [309]. Its activation can induce cell proliferation in the liver or prohibit apoptosis through several ways [310]. This receptor is involved in regulating cell growth in various cancers, such as colon, ovarian, prostate, endometrial, and osteosarcoma cancer, and have also been shown to regulate metastasis in lung cancer cells [311,312,313,314,315]. In the case of NSCLC, PXR expression was reported to be upregulated in vitro, and the expressions of cytochrome P450 2C8 (CYP2C8) and P-glycoprotein (P-gp) in NSCLC cell lines were increased after exposure to SR12813, an agonist of PXR. Results suggest that PXR expression has a significant impact on NSCLC cell resistance to Taxol by upregulating P-gp and CYP2C8 [316]. Its expression is found to be increased in NSCLC and can be targeted using a suitable agonist/antagonist in lung cancer cells.

### 2.11. Liver Receptor Homologue 1 (LRH-1)

LRH-1 belongs to the NR superfamily and is expressed primarily in tissues such as the liver, intestine, exocrine pancreas, and ovary. It is involved in regulating cholesterol transport, differentiation, steroidogenesis, bile acid homeostasis, and cancer progression [317]. Increased LRH-1 expression induces cell growth and survival, and any dysfunction can lead to a malignant condition. The LRH-1 structural organization consists of an NTD that lacks the ligand-independent AF-1 at the NH2 terminal, the highly conserved DBD (C domain) that directs the receptor to specific DNA sequences defined as HREs, and an LBD (E domain) containing a conserved ligand-dependent AF-2 motif that helps in coactivator interaction, and the D domain serving as hinge between DBD and LBD (Figure 3J) [317,318].

Earlier studies have shown that the LRH-1 overexpression can promote breast cancer resistance to chemotherapy [249,319], pancreatic cancer metastasis [320], colon cancer [321], NSCLC [118], and hepatoblastoma proliferation [322]. LRH-1 influences cancer cell growth, invasion, migration, and the epithelial–mesenchymal transition (EMT)/transformation, all of which are required for the tumor to remain malignant [323,324]. Therefore, LRH-1 appears to be a promising molecular target for cancer therapy. There has been very little research conducted on LRH-1 in lung cancer, suggesting possibilities of exploring LRH-1 agonists and antagonists in a step to hamper lung cancer cell growth and progression. In a study, LRH-1 was observed to be overexpressed in NSCLC cancer tissues compared to nearby normal lung tissues [325,326]. The results show that LRH-1 can be used for the prediction of NSCLC progression, metastasis, and poor prognosis, highlighting its potential as a new therapeutic target in NSCLC therapy [326]. In another study, LRH-1 was seen to be highly increased in NSCLC cell lines, and its knockdown considerably reduced cell proliferation and invasion, suggesting the involvement of LRH-1 in the regulation of NSCLC proliferation and invasion [327]. The role of microRNAs (miRNAs) has also been investigated in NSCLC, and they have also emerged as novel cancer regulators in recent years, encouraging cancer therapeutic strategies [328]. The miR-376c reduces NSCLC cell growth and invasion by targeting LRH-1, offering new insight into the possibility of developing anticancer medicines for NSCLC [327]. LRH-1 that is upregulated in lung cancer cells can be used as a novel therapeutic target using a specific small molecule for lung cancer therapy.

### 2.12. Glucocorticoid Receptor (GR)

GR is a ligand-activated TF and is ubiquitously found in almost all human tissues and organs including neural stem cells [329]. GR is transcribed from two highly homologous isoforms of the receptor, α and β. This receptor regulates various pathways in the body, mainly playing a role in anti-inflammatory pathways. It either binds to proteins encoding pro-inflammatory elements or to transcriptional factors which are pro-inflammatory [329]. It has diverse functions in various tissues in the body, from metabolism to growth to development to apoptosis [259]. The GR is made up of an NTD with a transcriptional AF-1, a DBD with two zinc-finger motifs that have a role in the recognition and binding to specific DNA sequences called glucocorticoid-responsive elements (GREs) in target genes, and a C-terminal LBD [330]. A flexible hinge region connects the DBD and LBD. The LBD has a ligand-dependent AF-2 that is bound to coregulators (Figure 3K) [331].

GRs have been examined for their anticancer effects on hematological malignancies [332]. A report has suggested higher levels of GR expression in NSCLC, which could be associated with better outcomes [333]. SCLC has been shown to have lower expression of GR than NSCLC [334]. The higher GR level has been observed in tumor cell lines and in vivo xenograft models, and mice models with a high GR content and lung tumor was found to respond to hydrocortisone or antiglucocorticoid, RU 38,486, by decreasing the tumor size [74]. In vitro studies have shown that treatment of transformed lung cell lines with dexamethasone (Dex) leads to the inhibition of cell proliferation and the reduction in the mRNA of both GR and K-ras [335]. Dex has growth inhibitory effects on the NSCLC cell lines in cultures, and these antiproliferative effects have been shown to be blocked by the use of antiglucocorticoid, RU-486 [336]. The Dex exerts growth inhibitory effects involving hypophosphorylation of Rb, reduced activity of CDK2 and 4, suppressed levels of cyclin D, E2F, and Myc, and enhanced levels of the CDK inhibitor p21(Cip1). Additionally, dexamethasone reduces the activity of ERK/MAPK, suggesting that glucocorticoid-induced cell cycle arrest and growth suppression could be mediated through modulation of the MAPK signaling system [337]. Dex has also been shown to induce 15-hydroxyprostaglandin dehydrogenase (15-PGDH), an enzyme in the catabolic pathway of prostaglandins [338]. This enzyme catalyzes the oxidation of prostaglandins 15(S)-hydroxyl group and is thought to be the primary enzyme in prostaglandin biological inactivation because 15-keto-prostaglandins drastically diminish biological activity [339]. The enzyme is found in abundance in lung tissues and may help to protect the cardiovascular system by eliminating vasoactive prostaglandins after they pass through the lungs [340]. If glucocorticoids reduce prostaglandin levels in tissues and the blood, they may increase prostaglandin catabolism and/or decrease prostaglandin biosynthesis [341]. A study suggested an additional mechanism for the anti-inflammatory action of these glucocorticoids [338]. Methylprednisolone, a synthetic glucocorticoid, has been shown to exert anticancer effects on lung cancer cell lines [342]. In a study, the expression of GR induced by infecting GR-expressing adenovirus was shown to inhibit tumor growth in a xenograft model and lead to a significant decrease in Bcl-2 and Bcl-xL transcripts, causing apoptosis. As a result, both in vitro and in vivo, GR expression is proapoptotic for human SCLCs, demonstrating that the lack of GR confers a survival advantage to SCLCs [343]. Findings have suggested that targeting GR by using its specific ligands can help to reduce tumor growth in lung cancer cells in vitro and in vivo.

### 2.13. Progesterone Receptor (PR)

PR is an NR that functions as a ligand-activated TF. The same gene generates two common isoforms (A and B) via distinct translational start sites; PR-B has the full length of the receptor, whereas PR-A is an N-terminally truncated form (missing the first 164 amino acids found in PR-B). The A and B isoforms can bind DNA at progesterone response elements (PREs) as homo- (A:A or B:B) or heterodimers (A:B) [344]. PR has a DBD, an LBD at the C-terminus, and several AFs and inhibitory functions (IFs) (Figure 3L). These AFs and IFs communicate with coregulators to either suppress or activate PR [345,346,347]. PR is found in a range of tissues, including the uterus, mammary gland, brain, pancreas, bone, ovary, testes, and lower urinary tract. PRA:PRB ratios in human cancers have been shown to be altered [348].

There are many studies available stating the deregulation of PR in cancer. In the case of lung cancer, alteration in the expression of PR subtypes has been observed to cause various changes affecting the cellular behavior and homeostasis. In a study, when compared to matching normal lung tissue, tumors exhibited lower expression levels of PR, which was reported to be an independent negative predictor of progression time (TTP) [115]. In a report, PR expression was shown to be significantly higher in SCLC compared with NSCLC [334]. It has been suggested that PR could be a potential prognostic factor in NSCLC, as PR levels were detected in 106 of 228 NSCLC patients (46.5%). The same study also showed a link of PR status to a better clinical outcome for the patients, and an analysis indicated it as an independent prognostic factor [107]. PR expression in another study was observed to be downregulated in tumor cell lines transplanted into a mouse [74]. Reports on lung cancer tissues and cell lines have showed reduced expression of PR [75,95,101]. In contrast, an increase in PR expression was also reported in 63% of tumors [104]. The deregulated expression of PR can be used as a prognostic marker in the case of lung cancer and considered as a target for treatment of lung cancer.

### 2.14. Thyroid Hormone Receptors (TRs)

TRs have subtypes, (TRα) and (TRβ), belonging to the steroid hormone receptor superfamily. They regularly associate with other NRs, such as RXR, RAR subtypes, and VDR, as homo- or heterodimers for proper function. The TR is made up of four basic domains that have been found to be evolutionarily conserved throughout the NR superfamily: a variable NTD containing a transactivation region AF-1, a central DBD consisting of two zinc fingers, a C-terminal LBD comprising dimerization interfaces and activation AF-2, and a linker or hinge region between the LBD and DBD that relates to DNA binding, AF and repression, ligand binding, and corepressor interactions (Figure 3M).

Studies have added support to the notion that the loss of TR functions may have a role in the development of human malignancies. Loss in THRB gene expression due to truncation/deletion of chromosome 3p has been observed in a wide range of cancers, including lung, melanoma, breast, renal cell, ovarian, head and neck, uterine cervical, and testicular tumors [349,350,351,352,353,354]. Studies have shown a significant increment in THRα gene expression in SCLC compared to NSCLC cases [355]. The same study demonstrated 91.6% of SCCs cases showing either intermediate or high expression, whereas 87.5% of NSCLC cases showed low THRα1 expression. High THRα1 expression was found to be linked to a shorter OS. These results suggest that THRα1 expression could be used not only as a prognosis marker, but also as a unique diagnostic additive tool for lung SCC, and it could also be considered as a potential therapeutic target for SCC in future [355]. Another report has shown that around 61% of SCLC and 48% of NSCLC cases lack TRβ1 expression, and 67% of SCLCs and 45% of NSCLCs carry TRβ1 promoter methylation. This decrease in the expression of THRB is due to promoter hypermethylation. Treatment with 5-aza-2-deoxycytidine (DAC) and/or trichostatin-A was seen to restore TRβ1 expression [356]. The levels of TRs are disturbed in lung cancer and can be explored for critical findings to establish it as a promising target for lung cancer therapy.

## 3. Biomarkers Using NRs in Lung Cancer

For the treatment of lung cancer, identifying prognostic biomarkers to establish therapeutic targets has been a valuable effort. Studies have shown that the genetic signature of NRs can serve as a prognostic biomarker for lung cancer, and many of NRs have been seen and thought to be druggable targets that can be therapeutically targeted for the treatment of various potential cancers [68,334,357,358,359,360,361,362,363]. In a study, the analysis of the expression of the 48 members of the NR superfamily in normal and lung cancer cell lines demonstrated that the NR gene expression can serve as a diagnostic biomarker that may be used for classifying lung cancer and predicting lung cancer incidence in smokers with 79% accuracy. Moreover, therapeutic potential of NR expression was shown for the prediction of ligand-dependent growth responses in lung cancer cells [334]. The same study suggested that profiling NR expression can predict the response ability of a cell to a ligand of a given receptor. A study was carried out to decipher the relevance between NR expression and the clinical outcome of the patients with NSCLCs, and reported two short heterodimer partners (SHPs) and PR as relevant markers. In addition, they suggested that the expression of either NR was valuable as a predictor of overall survival of patients [68]. According to another study, RAR methylation found in lung tissue can be employed as a marker for NSCLC diagnosis. It was reported in a study that overexpressed ERα in NSCLC patients correlates with smoking, and overexpression of both EGFR and ERα in NCLSC patients is associated with a poor prognosis and constitutes a useful prognostic factor [108]. In patients with NSCLC, LRH1 has been found to be a potential prognostic biomarker and predictor of metastasis [326].

## 4. Epigenetic Changes in NRs in Lung Cancer

Lung cancer possesses a series of genetic and epigenetic changes in the respiratory epithelium [143,364,365]. While somatic genetic abnormalities such as mutations and changes in copy number are widely known to have a role in oncogenesis, epigenetic perturbations are more common than somatic mutations in lung cancer [366,367,368]. Tumor suppressor gene silencing by promoter methylation, also known as hypermethylation, is a hallmark of lung cancer and is an early step in the carcinogenic process [369,370]. The promoter methylation of certain tumor suppressor genes, as well as the total number of hypermethylated genes, rose with neoplastic advancement from hyperplasia to an ADK state [371,372]. Many of the tumor suppressor genes hypermethylated in lung cancer are also hypermethylated often in other forms of solid tumors [373]. Promoter methylation is widely detected in genes involved in critical processes, such as cell cycle regulation, growth, cell death, cellular adhesion, motility, and DNA repair, in premalignant and malignant stages [374]. A variety of epigenetic changes in the NRs have been studied in lung cancer cell lines and clinical subjects, and these changes directly influence the cellular behavior and various processes.

In a study on SCLC and NSCLC cell lines and tumor samples, a promoter for RARβ2 transcription was found to be hypermethylated, which repressed the expression of RARβ2, pushing cells to undergo malignant transformation. This promoter methylation was thought to be one of the mechanisms that can silence RARβ in lung cancer, and demethylation using different chemicals might be a useful approach in combating lung cancer [375,376,377,378,379,380,381,382]. The findings of a study suggest that the development of second primary lung cancers (SPLCs) in NSCLC was affected differently by hypermethylation of the RARβ2 promoter, and this was dependent on smoking status. In never-smokers and former smokers with NSCLC, combining retinoids and/or a demethylating agent may be useful in preventing SPLCs [383]. RARβ2 promoter methylation was shown to be frequently present in patients with a smoking history, and silencing of this tumor suppressor due to aberrant methylation is thought to play a critical role in the pathogenesis of NSCLC [384]. Methylation has been shown to be a mechanism that inactivates genes, and studies have demonstrated that demethylating agents such as DAC can help with re-expression of RARβ in silenced cell lines and tumor samples in lung cancer [385,386]. In another study, elevated levels of RARβ methylation were observed in two of the seven lung cancer cells, and treatment with 5-aza-2’-deoxycytidine restored retinoic acid responsiveness (29%). In RA-responsive cells, RA therapy elevated acetylation of histones H3 and H4 on the chromatin of the RARβ promoter. Only histone H4 acetylation increased in RA-refractory cells, regardless of whether or not the promoter was methylated. Thus, in lung cancer cell lines, reduction in histone H3 acetylation was consistently associated with RA refractoriness [385]. In human cancer cells, it has been shown that the introduction of the p21 (sdi1) gene through infection with adenovirus, which encodes a CDK inhibitor, increases RARβ mRNA and protein expression and promotes sensitivity to ATRA [387]. According to one study, RAR methylation found in lung tissues can be employed as a predictive marker for NSCLC diagnosis [381].

Another gene target for inactivation through promoter methylation is ER, which has found to be hypermethylated in lung tumors [388,389,390]. Studies have suggested that DNA-methylation-mediated declines in ER expression may lead to the development of NSCLC [390]. A study reporting ER methylation status showed that 36.4% and 20% of tumor samples were detected to have promoter methylation at ER in lung tumors of smokers and those who never smoked [391]. ER was shown to be methylated in many lung tumors induced by the particulate carcinogens carbon black (CB), diesel exhaust (DE), or beryllium metal [392]. The data from a study indicated that the ER promoter is hypermethylated in lung tumors but not adjacent normal lung tissues, suggesting this disturbed methylation pattern plays an important role in lung tumorigenesis [121]. In the same study, treatment of A549 lung cancer cell lines with 17-beta estradiol (E2) increased the expression of ER mRNA and eliminated ER hypermethylation.

A report on RXR has shown that RXRG methylation is increased in lung cancer, which downregulates RXRG, and this methylation-associated downregulation of the RXRG might play a crucial role in the development of NSCLC [393]. The methylation of histones H3K9 and H3K27 was reduced, and NR0B1 transcription was marginally increased after treatment with histone methylase inhibitors. This analysis revealed that CpG methylation within the NR0B1 promoter is not implicated in the in vivo regulation of NR0B1 expression, but hyperacetylation of histone H4 and demethylation of histones H3K9 and H3K27, as well as their interaction with the NR0B1 promoter, results in decondensed euchromatin for NR0B1 activation [333]. A study reported that 67% of SCLCs and 45% of NSCLCs have TRβ1 promoter methylation, and methylation of TRβ1 is significantly associated with reduced TRβ1 expression, which can be restored by the treatment with DAC and/or trichostatin-A in lung cancer cell lines [356].

## 5. Clinical Trials in NRs

A plethora of research has suggested that several NRs have the potential to be used as cancer therapeutics and demonstrated the use of agonists and antagonists as a promising approach to inhibit lung cancer growth and progression through influencing various genes and proteins functioning as a network in the cell, which may affect cellular behavior and processes. Several clinical trials involving various ligands specific to their respective NR have been conducted, and many are still ongoing. The effects of various agonists and antagonists have been studied through these clinical trials and concluded to be potential therapeutics to target cancer cells.

A phase II interventional clinical study (ClinicalTrials.gov Identifier: NCT01556191) involving Ful, an antiestrogen, has been completed, which aimed to study the effects of administration of Ful with EGFR-TKI on 379 female participants with advanced-stage NSCLC. Ful binds to the ER, inhibits it, and speeds up the degradation of the ER [394]. Another ongoing interventional study (ClinicalTrials.gov Identifier: NCT02852083), using a combination of chemicals and comprising 86 participants of all sexes in Phase II, aims to determine the safety and effectiveness of a combination modularized treatment of treosulfan, a PPAR agonist PIO, and clarithromycin in patients with SCC and NSCLC after platin failure. In one interventional study (ClinicalTrials.gov Identifier: NCT01199068) with the primary goal of investigating the pharmacokinetics of CS-7017, a selective PPARγ agonist of the TZD class, in combination with erlotinib, is being tested to evaluate the safety and tolerability of CS-7017 delivered orally twice a day in conjunction with erlotinib. This study has passed Phase I, having 15 participants with metastatic or unresectable locally advanced NSCLC who failed first-line therapy. In a study (ClinicalTrials.gov Identifier: NCT01199055) with the primary goal of investigating the pharmacokinetics of CS-7017 in combination with carboplatin and paclitaxel, as well as to assess the safety and tolerability of CS-7017 in combination with carboplatin and paclitaxel, 16 participants with metastatic or unresectable locally advanced NSCLC were given these drugs, and the effects were evaluated, and the results are being compiled. 

For advanced NSCLC, platinum-based chemotherapy is the conventional treatment, which, unfortunately, has a low survival and response rate (RR). New therapy options are attracting a lot of attention. The use of retinoids such as ATRA is one of these emerging therapies. A randomized Phase II trial (ClinicalTrials.gov Identifier: NCT01048645) involving 107 individuals with advanced NSCLC was conducted to determine whether combining cisplatin, paclitaxel, and ATRA with an acceptable toxicity profile improves RR and progression-free survival (PFS) in patients with advanced NSCLC, as well as its relationship with RARβ2 expression as a response biomarker. Several studies in preclinical models and human clinical trials have demonstrated the efficacy of retinoids in the treatment and prevention of cancer. The second-generation retinoid IRX4204 is a highly powerful and specific RXR activator. Because IRX4204 is substantially more potent and favors RXRs over RARs compared to bexarotene, a first-generation authorized RXR agonist medication, it could lead to fewer side effects and increased activity in clinical trials. Preclinical investigations of the combination of IRX4204 and erlotinib, as well as prior clinical studies on the combination of bexarotene and erlotinib, showed that the two drugs had at least additive benefits in the treatment of NSCLC. One such interventional study (ClinicalTrials.gov Identifier: NCT02991651) comprising 12 participants with the purpose of evaluating the safety and efficacy of IRX4204 in combination with other drugs, is being conducted and recruiting patients with its inclusion and exclusion criteria. RGX-104 stimulates LXR, which causes MDSCs and tumor blood vessels to be depleted. MDSCs prevent T-cells and other immune system cells from targeting tumors. An ongoing clinical trial (ClinicalTrials.gov Identifier: NCT02922764) with the enrollment of 135 participants aimed to determine the efficacy of RGX-104 and explore RGX-104 as a single agent and as combination therapy in patients with advanced solid malignancies, including NSCLC. According to the findings, extensive Dex therapy promotes irreversible cell cycle blockage and a senescence phenotype in GR-overexpressing lung ADK cell populations through persistent activation of the p27Kip1 gene. In the first stage, three Dex escalation cohorts will be examined progressively from Cohort A to Cohort C until > two patients show a 3-deoxy-3-18F-fluorothymidine (FLT) positron emission tomography (PET) response (i.e., a 30% reduction in FLT-PET uptake in at least one target lesion). The other goal is to see how Dex affects pembrolizumab response rates in terms of overall response rate (ORR). According to early findings, Dex is expected to promote tumor senescence in at least one lesion in 60% of patients and increase overall response to pembrolizumab by 33%.

Despite the fact that several clinical trials have been conducted to better understand the effects of NRs and their druggability in other cancers, limited emphasis has been paid to find lung cancer therapies. As a result, more randomized, multicenter trials are needed to develop therapies targeting NRs in lung malignancies.

**Table 1 pharmaceuticals-15-00624-t001:** Nuclear receptor (NR) expression in lung cancer tissues and cell lines.

Nuclear Receptors (NRs)	In Vitro/In Vivo/Clinical	Models/Cell Lines/Tissue	Expression (Down-/Upregulation)	Reference
AR	In vivo	Athymic nude mice	Down	[74]
	In vitro	NSCLC cell lines	Down	[75]
	In vivo	Lewis’s lung cancer C57BL/6 mice	Up	[76]
COUP-TF	In vitro	Calu-6, H460, H596, SK-MES-1, H661	Up	[395]
ER	In vivo	Athymic nude mice	Down	[74]
	Clinical	Lung cancer tissues	Down	[101]
	In vitro	NSCLC cell lines	Down	[75]
	Clinical	NSCLC tissues	Up	[95]
	Clinical	Lung carcinoma tissues	Up	[102]
	Clinical	NSCLC tissues	Up	[103]
ERRα	Clinical	NSCLC tissues	Up	[298]
	In vitro	A549, H1793, H1395, H358	Up	[298]
ERα	Clinical	NSCLC tissues	Up	[104]
	In vitro	91T, 784T, 54T	Up	[106]
	In vitro	128-88T, H23, A549, Calu-6	Down	[106]
	Clinical	NSCLC tissues	Up	[107]
	Clinical	NSCLC specimens	Up	[105]
	Clinical	NSCLC specimens	Up	[108]
	In vivo	Xenograft SCID mice	Up	[106]
	Clinical	NSCLC tissues	Down	[109]
	Clinical	Lung cancer tissues	Down	[110]
	Clinical	NSCLC tissues	Down	[112]
	In vivo	Lewis’s lung cancer C57BL/6 mice	Up	[76]
ERβ	Clinical	NSCLC tissues	Up	[104]
	Clinical	NSCLC tissues	Up	[107]
	Clinical	NSCLC specimens	Down	[105]
	In vivo	Xenograft SCID mice	Up	[106]
	Clinical	NSCLC tissues	Up	[109]
	Clinical	Lung cancer tissues	Up	[110]
	Clinical	NSCLC tissues	Up	[112]
	Clinical	NSCLC specimens	Up	[111]
	Clinical	Lung cancer specimens	Up	[100]
	In vitro	ERF-LC-OK, PC-3, DMS114, PC-6,	Up	[100]
	Clinical	Lung cancer tissues	Up	[113]
	Clinical	Primary lung tumor tissues	Up	[115]
	Clinical	NSCLC specimens	Up	[116]
	Clinical	NSCLC tissues	Up	[117]
	In vitro	A549	Up	[117]
	In vivo	Lewis’s lung cancer C57BL/6 mice	Up	[76]
	Clinical	Primary NSCLC samples	Up	[118]
	Clinical	NSCLC tissues	Up	[114]
FXR	Clinical	NSCLC tissues	Up	[308]
	In vitro	H1975 and H1299	Up	[308]
GR	In vivo	Athymic nude mice	Up	[74]
	Clinical	Lung cancer tissues	Up	[101]
	Clinical	Lung neoplastic tissues	Up	[396]
	In vitro	NSCLC cell lines	Up	[397]
	In vitro	EPLC-32M1, H157, EPLC-272H, U-1752, A-549, H596, LCLC-97TM1	Up	[336]
	Clinical	NSCLC specimens	Up	[398]
LRH1	Clinical	NSCLC tissues	Up	[326]
	In vitro	A549, NCI-H157, H1299, SK-MES-1	Up	[327]
LRH-1 (Nr5a2)	Clinical	Lung cancer tissues	Up	[325]
MR	Clinical	Lung cancer tissues	Up	[399]
Nur77	In vitro	H520, H292	Up	[395]
PPARβ/δ	Clinical	Lung cancer tissues	Up	[155]
PPARγ	In vitro	A549, LTEP-P	Up	[147]
	Clinical	Lung cancer tissues	Down	[148]
	Clinical	NSCLC specimens	Up	[149]
	Clinical	NSCLC tissues	Down	[150]
	In vitro	H441, A549, H322, H1944	Up	[400]
	Clinical	NSCLC tissue	Up	[400]
	In vitro	H841, A549, PC14	Up	[151]
	Clinical	NSCLC specimens	Up	[152]
	In vivo	NNK-induced A/J mouse tumor	Up	[153]
	In vitro	Precancerous human bronchial epithelial cells (HBECs)	Up	[154]
PR	In vivo	Athymic nude mice	Down	[74]
	Clinical	Lung cancer tissues	Down	[101]
	Clinical	NSCLC tissues	Down	[95]
	Clinical	NSCLC tissues	Up	[104]
	Clinical	NSCLC tissues	Up	[107]
	In vitro	A549, LCSC#2, 1–87	Up	[107]
	Clinical	Primary lung tumor tissues	Down	[115]
	Clinical	NSCLC tissues	Down	[401]
PRα	Clinical	NSCLC specimens	Up	[402]
PXR	In vitro	A549, NCI-H358, HCC827, H1650, H1299	Up	[316]
RARα	Clinical	Lung cancerous tissue	Up	[207]
	In vitro	Calu-1, H647, Al188	Up	[222]
	In vitro	EBC-1	Up	[208]
	In vitro	Calu-1	Up	[209]
	In vitro	Rat tracheobronchial epithelial cell line SPOC-1	Up	[210]
	Clinical	NSCLC tissues	Down	[211]
	Clinical	NSCLC tissues	Down	[212]
	Clinical	NSCLC specimens	Down	[213]
	Clinical	NSCLC specimens	Down	[214]
RARβ	In vitro	H125, SK-MES-1, A1188, H596	Down	[222]
	In vitro	Calu-1	Down	[209]
	Clinical	NSCLC tissues	Down	[211]
	Clinical	Lung cancer tissues	Down	[216]
	In vitro	BEAS-2B-R1	Down	[216]
	Clinical	NSCLC tissues	Down	[212]
	Clinical	NSCLC specimens	Up	[217]
	Clinical	NSCLC specimens	Down	[213]
	In vitro	Calu-6, H146, H661, SK-MES-1	Down	[205]
	In vitro	Calu-3, H292	Up	[205]
	Clinical	Lung cancerous tissue	Down	[207]
	Clinical	Lung cancerous tissue	Down	[206]
	Clinical	Bronchial biopsy specimens	Down	[218]
RARγ	In vitro	H125, H647, SK-LU-1, H292	Up	[222]
	In vitro	Calu-1	Up	[209]
	In vitro	SPOC-1	Up	[210]
	Clinical	NSCLC tissues	Down	[212]
	Clinical	Lung cancerous tissue	Down	[207]
	Clinical	Lung cancerous tissue	Down	[206]
	In vitro	SK-MES-1, Calu-1, H157, H226, H460, H1792, H1648, H1944	Up	[221]
RORC2	Clinical	Lung cancer tissues	Up	[403]
RXR	Clinical	NSCLC specimens	Down	[214]
RXRα	Clinical	Lung cancerous tissue	Up	[207]
	Clinical	NSCLC tissues	Down	[212]
	Clinical	NSCLC tissues	Down	[245]
RXRβ	In vitro	SPOC-1	Up	[210]
	Clinical	NSCLC tissues	Down	[212]
	Clinical	NSCLC tissues	Down	[245]
	Clinical	Lung cancerous tissue	Down	[207]
	Clinical	Lung cancerous tissue	Down	[206]
RXRγ	Clinical	NSCLC tissues	Down	[212]
	Clinical	NSCLC tissues	Down	[245]
	Clinical	NSCLC specimens	Down	[214]
THRα1	Clinical	NSCLC tissues	Up	[355]
TR3 (NR41A and Nur77)	In vitro	H460, Calu-6	Up	[404]
	Clinical	NSCLC tissues	Up	[405]
VDR	In vitro	NSCLC cell lines	Up	[257]
	In vitro	SCLC cell lines	Up	[257]
	In vitro	EBC-1	Up	[208]
	Clinical	NSCLC tissues	Up	[258]
	Clinical	NSCLC tissues	Up	[262]
	Clinical	Biopsy specimens	Up	[263]
	Clinical	Lung cancer tissues	Down	[264]

**Table 2 pharmaceuticals-15-00624-t002:** Mechanistic role of various nuclear receptors in lung cancer in the presence of their agonists/antagonists.

Nuclear Receptors (NRs)	In Vitro/In Vivo/Clinical	Models/Cell Lines	Agonist/Antagonist	Results	Reference
AR	In vitro	86M1, 21H, 16HV, 24H	Testosterone (Ac)	↑ cell growth	[77]
	In vitro	H526, 86M1, 21H	Flutamide or Cyproterone acetate (In)	↓ cell growth	[77]
	In vitro	H1355	5α-Dihydrotestosterone (DHT) (Ac)	↑ cell growth	[78]
	In vitro	A549	DHT (Ac)	↑ cell growth, ↑ CD1,	[79]
ER	In vitro	H460	Tamoxifen + paclitaxel (In)	↓ cell growth, ↑ apoptosis	[406]
	In vitro	H23	β-Estradiol (Ac)	↑ cell growth	[106]
	In vitro	H23	Estradiol-17β (E2β) (Ac)	↑ pMAPK, ↑ cell growth, ↑ VEGF	[120]
	In vitro	H23	Faslodex (In)	↓ pMAPK, ↓ cell growth, ↓ VEGF	[120]
	In vitro	A549	17β-estradiol (E2) (Ac)	Restore ER mRNA, ↑ acetylated histone 3 and histone 4	[121]
	In vitro	201T, 273T, A549	E2 (Ac)	↑ cell proliferation, ↑ pMAPK, ↑ pEGFR,	[122]
	In vitro	201T, 273T, A549	Fulvestrant + gefitinib	↓ cell growth, ↑ apoptosis	[122]
	In vitro	NSCLC cells	Estrogen (Ac)	↑ ERβ, ↑ metastasis, ↑ MMP-2, ↑ TLR4	[123]
	In vitro	PC9, Hcc827	β-estradiol (Ac)	↑ ERβ1	[118]
	In vitro	H23, A549	β-estradiol (Ac)	↑ cell proliferation	[106]
	In vivo	Xenograft SCID mice	β-estradiol (Ac)	↑ cell proliferation	[106]
	In vivo	Xenograft SCID mice	ICI 182,780 (In) or β-estradiol + ICI 182,780	↓ cell proliferation	[106]
	In vitro	H23	siRNA-ERα or siRNA-ERβ	↓ ERα/ERβ, ↓ cell growth	[124]
	In vitro	A549, H1650	Tamoxifen (In- ER) + gefitinib	↓ cell growth, ↑ apoptosis, ↑ ERβ	[125]
ERRα	In vitro	A549, H1793	siRNA-ERRα (In) or XCT-790 (In)	↓ proliferation, ↓ migration, ↓ invasion, ↓ fibronectin (FN), ↓ vimentin, ↓ MMP-2, ↓ IL-6	[298]
ERβ	In vitro	Calu-6, 201T	ShRNA- ERβ	↓ cell growth, ↑ apoptosis	[119]
	In vitro	A549	ShRNA- ERβ (In)	↓ pERK, ↓ MMP-2, ↓ MMP-9, cell ↓ proliferation, ↓ invasion	[117]
	In vivo	Lung metastatic mouse model	ShRNA- ERβ (In)	↓ tumor growth, ↓ metastasis	[117]
	In vitro	A549, H1793	Estrogen (E2) (Ac)	↑ cell growth, ↑ ERβ, ↑ IL6, ↑ migration	[114]
	In vitro	A549, H1793	Fulvestrant (Ful) (In)	↓ cell growth, ↓ IL6,	[114]
	In vivo	Urethane-induced mouse model	Estrogen (E2) (Ac)	↑ tumor, ↑ ERβ, ↑ IL6, ↑ p-↑ p38MAPK, ↑ p-AKT, ↑ p-Stat3	[114]
FXR	In vitro	H1975, H1299	Z-guggulsterone (In)	↓ proliferation, ↓ CD1, ↓ CDK2, ↓ CDK4, ↓ CDK6, ↓ p-Rb	[308]
	In vitro	H1975, H1299	siRNA-FXR (In)	↓ proliferation, ↓ CD1, ↓ pRb	[308]
	In vivo	NSCLC xenograft	ShRNA-FXR (In)	↓ tumor growth,	[308]
GR	In vivo	Athymic nude mice	Hydrocortisone (Ac) or RU 38,486 (In)	↓ tumor size	[74]
	In vitro	C10	Dexamethasone (Dex) (Ac)	↓ cell proliferation, ↓ GR, ↓ K-RAS	[335]
	In vitro	A5, LM2 cells	Dex (Ac)	↓ GR	[335]
	In vitro	32M1, H157, A549, 97TM1	Dex (Ac)	↓ cell growth	[336]
	In vitro	32M1, H157, A549, 97TM1	RU-486 (In)	↑ cell growth	[336]
	In vitro	H82, H345, H510A, N592, H2081	Methylprednisolone (MP) (Ac)	↓ cell growth, ↑ apoptosis	[342]
	In vitro	A549, Calu-1 cells	Dex (Ac)	↓ pRB, ↓ CDK2, CDK4, ↓ cyclin D, ↓ E2F, ↓ Myc, ↑ p21(Cip1), ↓ ERK/MAPK	[337]
	In vitro	A549	Dex (Ac)	↑ 15-PGDH	[338]
	In vivo	SCLC xenograft mice	Infection with GR-expressing adenovirus	↓ tumor growth, ↓ Bcl-2, ↓ Bcl-xL, ↑ apoptosis,	[343]
LXR	In vitro	A549, HCC827-8-1	T0901317 (Ac) + gefitinib	↓ migration, ↓ invasion, ↓ MMP9, ↑ E-cadherin	[277]
	In vivo	BALB/c nude mice	T0901317 (Ac) + gefitinib	↓ migration, ↓ MMP9,	[277]
	In vivo	Homograft murine model	GW3965 or RGX-104 (Ac)	↓ myeloid-derived suppressor cell (MDSC)	[279]
	In vitro	A549	T0901317 (Ac)	↓ migration, ↓ invasion, ↓ MMP-9, ↓ NF-κB/MMP-9	[278]
	In vitro	HCC827/GR-8-2	GW3965 + gefitinib	↓ cell proliferation, ↑ apoptosis, ↓ NF-κB	[281]
	In vitro	H827-7-2, H827-7-4	GW3965 + gefitinib	↓ cell proliferation, ↑ apoptosis, ↓ pAKT, ↓ pNF-κB	[287]
LXRα	In vitro	HCC827-GR, PC9-GR	T0901317	↓ proliferation, ↑ LXRα, ↑ ABCA1	[288]
	In vitro	HCC827-GR, PC9-GR	siRNA- LXRα (In)	↓ LXRα, ↓ ABCA1, ↑ proliferation	[288]
NR0B1	In vitro	A549	siRNA- NR0B1 (In)	↓ Bcl-2, ↓ MMP-2	[407]
PPARα	In vivo	TC-1 lung tumor-mice model	AVE8134, Wyeth-14,643, Bezafibrate (Ac)	↓ EET, ↑ 11-HETE, ↑ proliferation, ↑ angiogenesis, ↑ AKT/ERK, ↑	[408]
PPARβ/δ	In vitro	A549, H23, H157	GW501516 (Ac)	↑ cell growth, ↓ apoptosis, ↑ pAkt, ↑ PDK1, ↓ PTEN, ↑ Bcl-xL, and COX-2	[155]
	In vitro	A549, H1838	GW0742 or GW501516 (Ac)	↑ Angptl4	[156]
	In vitro	H157, H1838	GW501516 (Ac)	↑ EP4, ↑ cell proliferation	[157]
PPARγ	In vitro	H441, H358	Ciglitizone or 15d-PGJ2 (Ac)	↓ cell growth, ↑ cell death, ↑ HTI_56_	[400]
	In vitro	H157, H322, H520, H522, H1299, H1334, H1944, A549	Ciglitizone or 15d-PGJ2 (Ac)	↓ cell growth, ↑ cell death	[400]
	In vitro	H441, H358, H322	Ciglitizone or 15d-PGJ2 (Ac)	↑ gelsolin, ↑ Mad, ↑ p21, ↑ PPARγ, ↑ HTI_56_ ↓ MUC1, ↓ SP-A, ↓ CC10	[400]
	In vitro	H157, H1299	Ciglitizone (Ac)	↓ MMP-2,	[400]
	In vitro	H441, H358	Ciglitizone(Ac)	↓ cyclin D1, ↓ pRb	[400]
	In vitro	H358	15d-PGJ2 (Ac)	↓ cyclin D1, ↓ pRb	[400]
	In vitro	H841, A549, PC14	Troglitazone (Tro) or 15d-PGJ2 (Ac)	↓ cell growth, ↑ cell death	[151]
	In vitro	A549, H345, N417	15d-PGJ2 (Ac)	↓ cell growth, ↑ apoptosis	[409]
	In vitro	A549, N417	Ciglitazone (Ac)	↓ cell growth	[409]
	In vitro	A549, LTEP-P	15d-PGJ(2) or ciglitazone (Ac)	↓ cell growth, ↑ apoptosis, ↑ Caspase-3, ↑ bax, ↑ bcl-2	[158]
	In vivo/In vitro	Nude mice-A549 cells	Ciglitazone (Ac)	↓ cell growth, ↑ PPARγ, ↓ cyclin D1, ↑ P21	[410]
	In vitro	A549, LTEP-P	15d-PGJ2 or ciglitazone (Ac)	↓ cell viability, ↑ apoptosis	[147]
	In vitro	A549	Tro (Ac)	↑ PPARγ activity, ↓ cell growth, ↓ cyclins D and E, ↓ Erk1/2	[149]
	In vivo	A549-tumor-bearing SCID mice	Tro or Pio (Ac)	↓ tumor	[149]
	In vitro	H1838, H2106	15d-PGJ2, rosiglitazone (BRL49653) or Tro (Ac)	↓ fibronectin (Fn), ↓ pCREB, ↓ Sp1	[164]
	In vitro	H1838, H2106	GW-9662 (In)	↑ fibronectin (Fn)	[164]
	In vitro	H1838, H2106	siRNA- PPARγ (In)	↑ fibronectin (Fn)	[164]
	In vitro	A549, H2122	ciglitazone, PGA1, or 15-deoxy-12,14-PGJ2 (Ac)	↓ cell growth	[165]
	In vitro	H2122	ciglitazone	↑ E-cadherin, ↓ cell growth	[165]
	In vivo/In vitro	Nude rats-H2122-PPAR(gamma) cells	Implantation of H2122- PPARγ cells into the lungs of nude rats	↓ cell growth, ↓ metastasis,	[166]
	In vitro	SQ-5, EBC-1, ABC-1, RERF-LC-OK	Tro (Ac)	↓ cell growth, ↑ apoptosis, ↑ GADD153	[167]
	In vitro	H460, H1299, H661	Thalidomide (Ac)	↑ PPARγ, ↓ NFκB, ↑ apoptosis, ↓ growth-related oncogene (GRO), ↓ epithelial cell derived-neutrophil activating peptide-78 (ENA-78), ↓ angiotensin, ↓ IL-8, ↓ COX-2	[168]
	In vivo	Nude mice-NCI-H1299 cells	Thalidomide (Ac)	↓ tumor growth, ↑ PPARγ	[168]
	In vitro	H23, CRL-2066	Tro (Ac)	↓ cell growth, ↑ apoptosis, ↑ PPARγ, ↓ Bcl-w, ↓ Bcl-2, ↑ ERK1/2, ↑ p38, ↓ SAPK/JNK	[169]
	In vitro	H1838, H2106	GW1929, PGJ2, ciglitazone, Tro or rosiglitazone	↓ cell growth, ↑ pErk, ↓ EP2,	[170]
	In vitro	H345, H2081, H1838, H2106	PGJ2 or ciglitazone	↓ cell growth, ↑ apoptosis, ↑ p21, ↓ cyclin D1	[171]
	In vitro	A549	Rosiglitazone	↓ cell growth, ↑ PPARγ, ↑ PTEN,	[173]
	In vitro	H23	Troglitazone	↑ ERK1/2, ↑ apoptosis, ↑ Akt	[172]
	In vitro	H23	siRNA- PPARγ (In)	↓ ERK1/2, ↓apoptosis,	[172]
	In vitro	H2122	SR 202 (In)	↓ PPARγ, ↑ cell growth, ↓ E-cadherin	[174]
	In vitro	A549, H157, H460, H1792	Tro, cigolitazone and GW1929	↓ FLIP_L_, ↓ FLIP_S_, ↑ DR5,	[411]
	In vivo	A549, H460	15d-PGJ2	↓ cell growth, ↑ apoptosis, ↑ PPARγ, ↑ caspase3, ↓ Cyclin D1	[175]
	In vivo	A549, H460	Docetaxel	↓ cell growth, ↑ apoptosis, ↑ caspase3, ↓ Cyclin D1	[175]
	In vivo	A549, H460	15d-PGJ2 + docetaxel	↓ cell growth, ↑ apoptosis, ↑ PPARγ, ↑ caspase3, ↓ Cyclin D11	[175]
	In vivo	Athymic nu/nu mice- A549 and H460 cells	15d-PGJ2	↓ tumor volume, ↓ PGE2, ↑ PPARγ, ↑ caspase3, ↑ caspase8, ↓ Cyclin D1, ↑ BAD, ↓ Bcl2	[175]
	In vivo	Athymic nu/nu mice- A549 and H460 cells	Docetaxel	↓ tumor volume, ↓ PGE2, ↑ PPARγ, ↑ caspase3, ↑ caspase8, ↑ caspase9, ↓ Cyclin D1, ↑ BAD, ↓ Bcl2, ↑ APAF1, ↑ BBC3, ↑ p53	[175]
	In vivo	Athymic nu/nu mice- A549 and H460 cells	15d-PGJ2 + docetaxel	↓ tumor volume, ↓ PGE2, ↑ PPARγ, ↑ caspase3, ↑ caspase8, ↑ caspase9, ↓ Cyclin D1, ↑ BAD, ↓ Bcl2, ↑ APAF1, ↑ BBC3, ↑ p53	[175]
	In vitro	CL1-0, A549	Tro + Aspirin	↓ cell growth, ↓ Cdk2, ↓ E2F-1, ↓ cyclin B1, ↓ cyclin D3, ↓ pRB, ↑ apoptosis, ↓ PI3K/Akt, ↑ p27, ↓ pRac1	[412]
	In vitro	A549	KR-62980 or rosiglitazone	↓ cell growth, ↑ apoptosis, ↑ ROS, ↑ proline oxidase (POX),	[176]
	In vitro	A549	Bisphenol A diglycidyl ether (In)	↓ apoptosis	[176]
	In vitro	RERF-LC-AI, SK-MES-1, PC-14, A549	Tro or ciglitazone	↑ VEGF, ↑ neuropilin-1	[177]
	In vitro	RERF-LC-AI, SK-MES-1, PC-14, A549	GW9662 (In)	↓ VEGF, ↓ neuropilin-1	[177]
	In vitro	A549	Tro and rosiglitazone	↓ TGF-β-induced EMT, ↓ vimentin, ↓ N-cadherin, ↓ fibronectin, ↓ migration, ↓ invasion	[413]
	In vivo	Adenocarcinoma and SCC A/J mice	Pioglitazone	↓ tumor, ↑ apoptosis	[414]
	In vitro	H1838, H2106, A549	Rosiglitazone (Ac)	↓ cell proliferation, ↓ alpha4 nicotinic acetylcholine receptor (nAChR), ↑ p38 MAPK, ↑ ERK 1/2, ↓ pAkt, ↑ p53	[415]
	In vivo	A549-induced nude mice tumor	Ciglitazone (Ac)	↓ cell proliferation, ↑ PPARγ, ↓ cyclin D1, ↑ P21	[180]
	In vitro	A549R, H460R	CB13 (Ac)	↓ cell viability, ↑ LDH, ↑ caspase-3, ↑ caspase-9, ↑ ROS, ↑ apoptosis	[181]
	In vitro	A549, H460	CB11 (Ac)	↓ cell viability, ↑ apoptosis, ↑ p-ATM, ↑ p-chk2, ↑ p-p53, ↑ GADD45α, ↑ LDH, ↑ caspase-3, ↑ PPARγ	[182]
	In vivo	A549 Xenograft mice	CB11 (Ac)	↑ apoptosis, ↓ tumor volume, ↓ EMT, ↑ caspase-3, ↑ caspase-9, ↑ PPARγ	[182]
	In vitro	H1299, H460	PIO (Ac)	↓ cell proliferation, ↑ apoptosis, ↑ caspase-3, ↓ Myc, ↓ R-Ras, ↓ MAPK6, ↓ MAP3K8, ↓ Bcl-2, ↓ PCNA, ↓ laminin, ↑ CASP5, ↑ CASP4, ↑ CFLAR, ↑ PAWR, ↓ TGFβR1, ↓ SMAD3, ↓ pEGFR, ↓ pAKT, ↓ pMAPK	[179]
	In vitro	A549	Bavachinin (BNN) (Ac)	↓ cell viability, ↑ ROS, ↑ apoptosis	[416]
	In vitro	SCC-15	Les-2194 (Ac)	↑ ROS, ↓ Ki67	[183]
	In vitro	SCC-15, A549	Les-3377 (Ac)	↑ ROS	[183]
	In vitro	SCC-15	Les-3640 (Ac)	↑ ROS, ↑ caspase-3, ↑ PPARγ	[183]
	In vitro	A549	Rosiglitazone (Ac)	↑ cell death, ↓ pAKT	[417]
	In vitro	H460, A549	PPZ023 (Ac)	↓ cell growth, ↑ LDH, ↑ apoptosis, ↑ PPARɣ, ↑ caspase-3, ↑ caspase-8, ↑ caspase-9	[184]
	In vitro	H1993	Thiazolidinedione (TZD) (Ac)	↓ ALDH1A3	[418]
	In vitro	HCC827-GR, PC9-GR	Efatutazone (Ac)	↓ proliferation, ↑ PPARγ, ↑ LXRα, ↑ ABCA1	[288]
	In vitro	Precancerous human bronchial epithelial cells (HBECs)	TZD (Ac)	↓ COX2, ↓ cell growth, ↓ clonogenicity, ↓ cell migration	[154]
PPARγ/RXR	In vitro	Calu-6	ciglitazone (Ac- PPARγ) + SR11237 (Ac-RXR)	↓ cell growth, ↑ RAR-β	[185]
	In vitro	Calu-6	ciglitazone (Ac- PPARγ) and SR11237 (Ac-RXR) + bisphenol A diglycidyl ether (In- PPARγ)	↓ RAR-β	[185]
PPARδ	In vitro	A549	L-165041 (Ac)	↓ cell growth, ↓ cyclin D, ↓ PCNA	[186]
	In vitro	A549	SR13904 (In)	↓ cell proliferation, ↑ apoptosis	[187]
PR	In vitro	A549, LCSC#2, 1-87	Progesterone (Ac)	↓ cell proliferation	[107]
	In vitro	A549, LCSC#2, 1-87	RU 38,486 (In)	↑ cell proliferation	[107]
	In vivo/In vitro	Athymic nude mice cells (A549, 1-87, and LCSC#2)	Progesterone (Ac)	↓ tumor volume, ↑ p21, ↑ p27, ↓ cyclin A, ↓ cyclin E, ↓ Ki67	[107]
PRα	In vitro	A549, PC-9, PC-9GR	P4/Org + gefitinib (Ac)	↓ proliferation, ↓ invasion, ↓ migration,	[402]
PXR	In vitro	A549	SR12813 (Ac)	↑ PXR, ↑ CYP2C8, ↑ P-gp	[316]
	In vitro	A549	siRNA- PXR (In)	↓ PXR, ↓ CYP2C8, ↓ P-gp	[316]
RAR	Clinical	Bronchial biopsy specimens	13-CRA (Ac)	↑ RAR-β	[218]
	Clinical	Bronchial brushing samples	13-CRA (Ac)	↑ RAR-β	[220]
	In vitro	H209-RAR-β	RA (Ac)	↓ cell growth, ↑ p27Kip1and ↓ L-myc, ↑ apoptosis, ↓ cdk2 kinase activity	[419]
	In vitro	H82	RA (Ac)	↓ cell growth, ↑ p27Kip1, ↑ RAR-β, ↓ cdk2 kinase activity	[419]
	In vitro	H460, Calu-1, SK-LU-1, A549, H69 cells	RA (Ac)	↑ RAR-β	[222]
	In vitro	EBC-1	RA (Ac)	↓ cell growth	[208]
	In vitro	H82	RA (Ac)	↓ cell growth	[215]
	In vivo	Male A/J mice	Isotretinoin (Ac)	↓ tumor multiplicity, ↑ RAR-α, ↑ RAR-β, ↑ RAR-γ	[223]
	In vitro	Calu-1	ATRA (Ac)	↑ cell growth (serum-free medium)	[209]
	In vitro	H292G, H358, H157, H1792, H226a	ATRA (Ac)	↓ cell growth	[209]
	In vitro	H345, H51 0	13-CRA (Ac)	↑ RAR-β, ↓ cell growth	[219]
	In vitro	CH27	RA (Ac)	↓ cell growth, ↑ p27(Kip1), ↓ Cdk3, ↓ p21(CIP1/Waf1), ↑ RAR-β, ↓ c-Myc, ↓ cyclin A/Cdk2 activity	[224]
	In vitro	SCLC	RA (Ac)	↑ gastrin-releasing peptide (GRP), ↑ cell growth	[225]
	In vitro	H460, SK-MES-1, H1792	CD437 (Ac)	↓ cell growth, ↑ apoptosis, ↑ c-Myc, ↑ ornithine decarboxylase (ODC), ↑ cdc25A	[226]
	In vitro	GLC82	RA or 4-HPR (Ac)	↓ cell growth, ↑ RAR-β2	[228]
	In vitro	H460	CD437 (Ac)	↓ cell growth, ↑ apoptosis, ↑ p53, ↑ p21, ↑ Bax, ↑ Killer/DR5	[227]
	In vitro	Rat tracheobronchial epithelial cell line SPOC-1	SRI-6751-84 (Ac) RARα-selective retinoid (Ro40-6055)	↑ transglutaminase (TGase II), ↑ apoptosis	[210]
	In vitro	Rat tracheobronchial epithelial cell line SPOC-1	RARα-antagonist Ro41-5253	↓ TGase II induced by RAR-selective retinoid, ↑ apoptosis	[210]
	In vitro	BZR-T33 ras transformed human bronchial epithelial cell line	ATRA (Ac)	↓ cell proliferation, ↑ RAR-γ,	[420]
	In vitro	H460	RA (Ac)	↑ EGFR, ↑ tumorigenicity	[229]
	In vitro	H460a	RA (Ac)	↑ EGFR, ↑ tumorigenicity	[230]
	In vitro	Calu-1, A549, H1792	ATRA (Ac)	↑ TIG3	[232]
	In vitro	A549, H1792, H157, H460	ATRA (Ac)	↑ TIG3, ↑ RAR-β	[232]
	In vitro	A549	beta-cryptoxanthin (Ac)	↓ cell growth, ↓ cyclin D1, ↓ cyclin E, ↑ p21, ↑ RAR-β	[233]
	In vitro	A549	ATRA (Ac)	↑ VEGF-C, ↑ VEGF-D, ↑ VEGFR3, ↓ RXRα,	[234]
	In vitro	A549	RA (Ac)	↓ invasiveness, ↑ CRABP	[421]
	In vitro	Calu-6	RA (Ac)	↑ RAR-β	[422]
	In vitro	Calu-1	RAR-α (Am80), RAR-β/γ (TTNN and SR3985) (Ac)	↑ cell growth (serum-free medium)	[209]
	In vitro	Calu-1	RAR-γ (CD2325 and SR11363) (Ac)	↓ cell growth (serum-free medium)	[209]
	In vitro	NSCLC cell lines	4HPR (Ac)	↓ cell growth, ↑ apoptosis	[235]
	In vitro	A549, H226, H1648, SK-MES-l	4-HPR (Ac)	↓ cell growth, ↓ Bcl-2, ↑ apoptosis	[235]
RAR/RAR-α	In vitro	Calu-1, H1792	AGN193109 (In-RAR) or Ro 41-5253 (In- RAR-α)	↓ TIG3	[232]
RAR/RXR	In vitro	Calu-6, H460	ATRA (Ac)	↓ cell growth, ↑ RAR-β, ↑ apoptosis	[423]
	In vivo	A/J mouse	9Cra (Ac)	↓ tumor multiplicity, ↑ RAR-β	[246]
RARγ	In vitro	SK-MES-1, Calu-1, H157, H226, H460, H1792, H1648, H1944	CD437 (Ac)	↓ cell growth, ↑ apoptosis	[221]
RXR	In vivo	A549 xenograft model	LGD1069 (Ac)	↓ cell growth, ↓ CD31, ↓ VEGF, ↓ JNK and ERK activation	[247]
	In vivo	Vinyl-carbamate-induced A/J mouse model	MSU42011 (Ac)	↓ tumor, ↑ CD8/CD4 ratio, ↑ CD25 T cells	[248]
	In vitro	A549	Bexarotene (Ac)	↑ PPARγ, ↑ PTEN, ↓ mTOR,	[249]
TR3 (NR41A and Nur77)	In vitro	H460, Calu-6	siRNA-TR3 (In)	↓ cell growth	[404]
	In vitro	A549, H460	siRNA-TR3 (In)	↓ cell growth, ↑ apoptosis, ↓ survivin, ↓ EGFR, ↓ bcl-2, ↓ c-myc, ↓ p70S6K, ↓ pS6RP, ↓ 4E-BP1, ↑ pAMPKα, ↑ sestrin	[405]
VDR	In vitro	EBC-1	calcitriol (Ac)	↓ cell growth	[208]
	In vivo/In vitro	C57BL/6 mice—green fluorescent protein-transfected Lewis lung carcinoma (LLC-GFP) cells	calcitriol (Ac)	↓ MMP-2, ↓ MMP-9, ↓ VEGF, ↓ parathyroid-hormone-related protein (PTHrP)	[265]
	In vitro	H460	Vitamin D	↓ histidine-rich calcium-binding protein (HRC), ↓ cell migration, ↓ proliferation, ↑ apoptosis	[264]

## 6. Conclusions and Future Perspectives

NRs have been found to have a critical role in lung cancer development and progression through significant research. New diagnostic and prognostic markers can be developed based on the expression and methylation pattern of a specific NR, which could be highly valuable in the early detection and better prognosis of lung and other malignancies. Through this review, we have presented the studies that show the involvement of several NRs in the development and progression of both SCLC and NSCLC, as well as the potential of using their agonists and antagonists to treat and prevent lung cancer (Figure 4). This study also focused on the roles of specific agonists and antagonists that can influence the expression of various types of NRs genes important for cell growth and maintenance, with the goal of creating possible lung cancer therapies. We have seen that several NRs are deregulated in lung cancer and the use of different specific agonists/antagonists has shown reliable results in the treatment of preclinical lung cancer models. NRs that are observed to be perturbed in lung cancer include AR, ER, RAR, RXR, PPAR, ERR, VDR, LXR, LRH-1, GR, PXR, PR, FXR, and TRs, and showed the potential to be used as biomarkers. Some of the NRs are observed to be upregulated in lung cancer cell lines and tissues as compared to their normal surrounding tissues, and they include ERβ, FXR, GR, LRH-1, MR, Nur77, PPARβ/δ, PPARγ, PXR, RORC2, THRα1, TR3, VDR, ERRα, while others, such as RARβ, RXRβ, RXRγ, are observed to be downregulated at a significant level. There are a few NRs, including AR, ERα, PR, RARα, RARγ, RXRα, which are differentially expressed in lung cancer tissues.

Lung carcinogenesis is thought to involve a network of complicated accessory proteins and signaling cascades that can be manipulated, using various agonists and antagonists targeting specific NRs. There have been few studies on primary lung tumor tissues that have supported that the deregulation of some NRs occurs at the early stage of lung cancer. In one such study, the expression of ERβ was observed to be higher in primary lung cancer tissues as compared to their normal surrounding tissues. As a result, numerous clinical trials have been conducted in order to demonstrate the efficacy of small molecules for lung cancer treatments. PPARγ is an NR which has extensively been studied in many cancers and shown to have antiproliferative effects when activated by its ligands, such as PIO, Tro, rosiglitazone, and ciglitazone. In lung cancer, its lower expression is associated with poor prognosis. Moreover, findings from various studies suggest that PPARγ can be targeted using a specific ligand in order to have control over cancer cell proliferation and inhibit tumor development and progression, and also could be established as a prognostic marker. VDR is also a promising candidate based on the previous studies where VDR was shown to play antiproliferative roles when activated by its specific ligand, calcitriol. VDR could also be considered as a factor for the treatment of lung cancer. FXR, which plays a role as an oncogene or tumor suppressor, can also be targeted using different strategies based on its expression in lung cancer. When overexpressed, FXR can be targeted using siRNA or other approaches to decrease its levels in order to abrogate the growth of cancer cells. The LRH-1 receptor, whose expression in lung cancer is decreased, was shown to inhibit tumor growth in pancreatic cancer. There has been very little research on LRH-1 in lung cancer, which suggests that LRH-1 agonists and antagonists could be used to inhibit lung cancer cell growth and tumor progression. Thus, LRH-1 could serve as a potential candidate for inhibiting the growth and progression of lung cancer cells.

Epigenetic changes may have a significant impact on gene expression in the cell, and many NRs have been observed to possess various epigenetic changes that hamper several cellular processes and cell behavior by affecting the expression of crucial genes. In lung cancer tissues and cell lines, the RARβ promoter has been observed to be hypermethylated, suppressing its expression and causing cells to undergo malignant transformation. Another gene that is found to be silenced due to abnormal methylation in NSCLC is ER, the aberrant expression of which can lead to development of lung cancer. Other established gene targets for inactivation by methylation as found by several studies are NR0B1, RARG, and TRβ1. Some demethylating agents, such as DAC and trichostatin-A, have shown promising results in the treatment of lung cancer cells involving hypermethylation of TRβ1 and RARβ. Employing RA therapy has also shown that acetylation of histones on the chromatin of the RARβ promoter can be enhanced to induce responsiveness of cells to RA. In order to combat lung cancer involving aberrant epigenetic changes, there is a requirement to explore possibilities of employing chemical agents such as the demethylating agents by treating cancer cell lines with suitable amounts. Despite the fact that much effort has been put into research into the mechanisms underlying lung cancer and therapies, more effective treatments that target NRs have the potential to be used as suitable approaches to prevent lung cancer incidences are still needed. From this perspective, gaining a thorough understanding of how NRs play a key role in lung cancer pathogenesis would aid in the development of novel therapeutic targets for improved lung cancer management.

## Figures and Tables

**Figure 1 pharmaceuticals-15-00624-f001:**
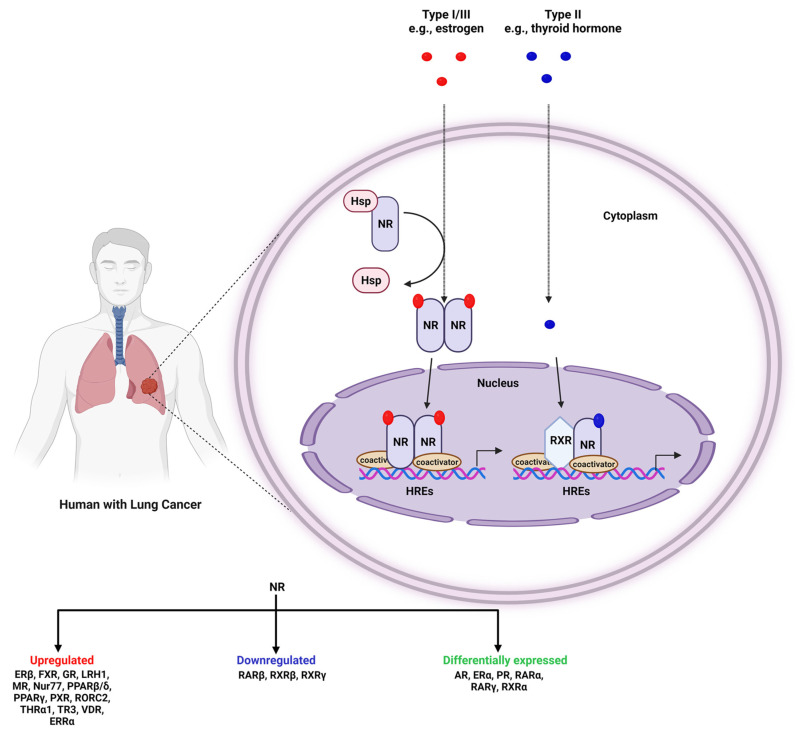
**NRs involved in lung tumorigenesis.** There are four major types of NRs: Type I/III receptors bind with the ligands in the cytoplasm, leading to their dissociation from the chaperons and subsequent nuclear translocation and transcription of active genes; in contrast, Type II receptors dimerize with the other NRs, and dissociate the corepressor through ligand binding, activating the NRs. Numerous studies have shown differential expression of various NRs in lung cancer.

**Figure 2 pharmaceuticals-15-00624-f002:**
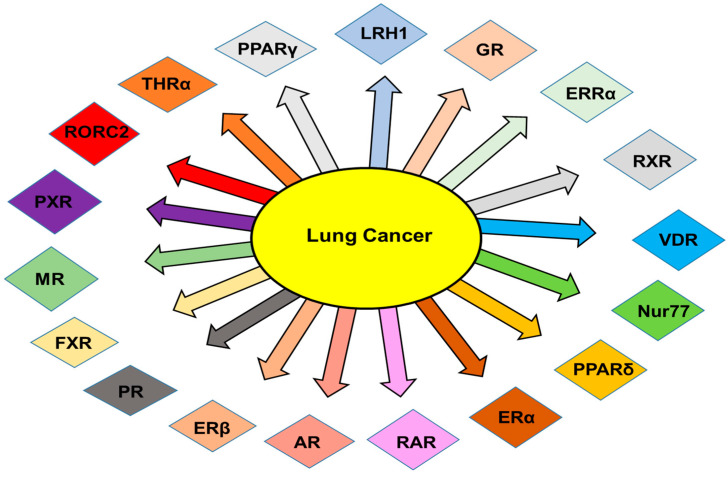
Different types of NRs implicated in lung cancer.

**Figure 4 pharmaceuticals-15-00624-f004:**
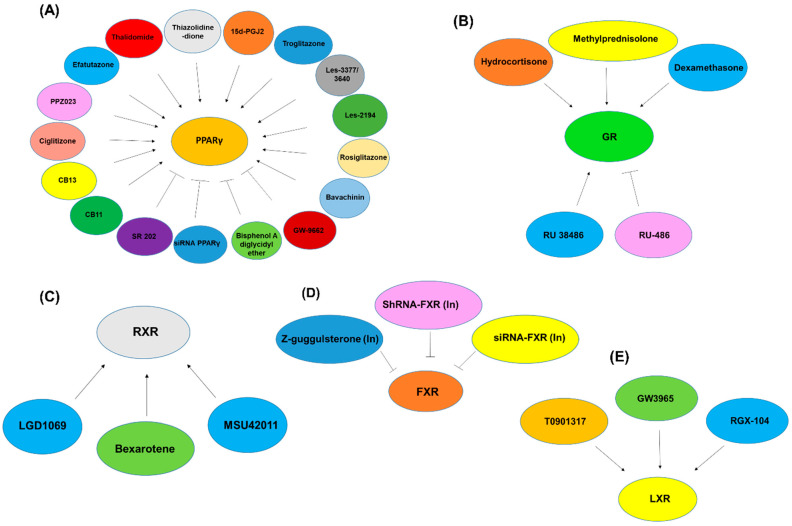
Nuclear receptors and their agonists/antagonists: (**A**) RARs, (**B**) GR, (**C**) RXR, (**D**) FXR, and (**E**) LXR.

## Data Availability

Data sharing is not applicable to this article as no new data were generated, and the article describes entirely theoretical research.

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
