# Peer review of "Targeting Nuclear Receptors in Lung Cancer—Novel Therapeutic Prospects"

_pharmaceuticals, 2022, doi:10.3390/ph15050624_

Round 1

Reviewer 1 Report

This is an interesting and comprehensive manuscript on nuclear receptors and lung cancer. However, there are still some suggestions that can help to improve the article further, which are explained below:

Comments:

  1. A brief paragraph on nuclear receptors (NRs) and other cancers should be included. Also, a short paragraph on targeting NRs in other cancers should be included if possible.
  2. Does dysregulation of NRs occur at the onset of lung cancer or only at an advanced stage?

3: Some of the NRs are overexpressed in lung cancer and others are downregulated? How do the authors account for the extent of NRs expression in lung cancer?

  1. There are some typos, grammatical errors and improper sentence structure here and there, so the English needs to be polished.

Author Response

We thank all reviewers for their valuable comments to greatly improved our manuscript. Below are our point-point-response to the comments

Reviewer 1:

This is an interesting and comprehensive manuscript on nuclear receptors and lung cancer. However, there are still some suggestions that can help to improve the article further, which are explained below:

Author’s Response: We are very much grateful to reviewer for the insightful comments which no doubt has helped increase the quality of the manuscript. Our point-by-point response for each comment is provided below

  1. A brief paragraph on nuclear receptors (NRs) and other cancers should be included. Also, a short paragraph on targeting NRs in other cancers should be included if possible.

Author’s Response: We thank the reviewer for the insightful comment. Based on the reviewer suggestion, we have included a brief section discussing the NRs and their role in targeting other cancers in the Introduction section.

  1. Does dysregulation of NRs occur at the onset of lung cancer or only at an advanced stage?

Author’s Response: We thank the reviewer for the comment. There have been few studies on primary lung tumor tissues that have supported that the deregulation of some NRs occurs at the early stage of lung cancer. In one such study, the expression of ERβ was observed to be higher in primary lung cancer tissues as compared to their normal surrounding tissues. As per the reviewer suggestion, we have discussed about the extent of deregulation of the NRs for lung cancer in the Conclusion section.

3: Some of the NRs are overexpressed in lung cancer and others are downregulated? How do the authors account for the extent of NRs expression in lung cancer?

Author’s Response: We thank the reviewer for the insightful comment. Various studies show that NRs have a wide range of expression in lung cancer. The function of NRs is dictated by their expression in lung tumours, and they can either promote or prevent tumorigenesis. For example, studies have indicated that RARs are differentially regulated in lung cancer. Retinoic acid and its derivatives (retinoids) are generally downregulated that bind to RARs possess differentiation, anti-proliferative, and pro-apoptotic, and are used as possible chemotherapeutic or chemopreventive drugs. Based on the reviewer suggestion, we have briefly discussed about the NRs expression in lung cancer at the Conclusion section.

  1. There are some typos, grammatical errors and improper sentence structure here and there, so the English needs to be polished.

Author’s Response: We thank the reviewer for the comment. We have checked and diligently modified the grammatical errors and errors in sentence construction. We have revised every line to correct all grammatical errors and rewrite/simplify complex and confusing sentences and have used the Grammarly Premium for language correction.

Best Regards,

Dr Alan Prem Kumar

Reviewer 2 Report

The authors  are comprehensively reviewing the knowledge about nuclear receptors (NRs) that may serve as targets for therapy of lung cancer, the major cause of fatalities worldwide.  In introduction they describe, in a very detailed manner, current knowledge about 13 important nuclear receptors and their association with lung cancer development and progression obtained through a significant  amount of research. Then they review relevant research on NRs as biomarkers for lung cancer, their epigenetic changes, and relevant clinical trials. The authors conclude that the research on NRs, especially AR, ER, RAR, RXR, PPAR, ERR, VDR, LXR, LRH-1, GR, 1182 PXR, PR, FXR, TRs and lung cancer shows the potential for using their agonists and antagonists to treat and prevent lung cancer.

Minor

Please avoid  statements as: „In this study, we have investigated the involvement of several NRs in the development“ ( line 1174). Rather write „In this review we  present data about involvement….“

Author Response

We thank all reviewers for their valuable comments to greatly improved our manuscript. Below are our point-point-response to the comments

Reviewer 2:

The authors are comprehensively reviewing the knowledge about nuclear receptors (NRs) that may serve as targets for therapy of lung cancer, the major cause of fatalities worldwide.  In introduction they describe, in a very detailed manner, current knowledge about 13 important nuclear receptors and their association with lung cancer development and progression obtained through a significant amount of research. Then they review relevant research on NRs as biomarkers for lung cancer, their epigenetic changes, and relevant clinical trials. The authors conclude that the research on NRs, especially AR, ER, RAR, RXR, PPAR, ERR, VDR, LXR, LRH-1, GR, 1182 PXR, PR, FXR, TRs and lung cancer shows the potential for using their agonists and antagonists to treat and prevent lung cancer.

Author’s Response: We thank the reviewer for the constructive comments.

  1. Please avoid statements as: „In this study, we have investigated the involvement of several NRs in the development “(line 1174). Rather write „In this review we present data about involvement…. “

Author’s Response: We thank the reviewer for the comment. Based on the reviewer suggestion, we have made changes in such statements.

Best Regards,

Dr Alan Prem Kumar

Reviewer 3 Report

Hello,

The review is in great shape. There are only few areas needs further needs further attention.

Minor:

  1. Please correct the font style in affiliation section.
  2. Figure 2 needs rework on the uniformity and spacing in the design of the arrows.
  3. The space margins between the lines/paragraphs needs attention.

Overall, great work with compilation of many literatures and summarize in the table format.

Author Response

We thank all reviewers for their valuable comments to greatly improved our manuscript. Below are our point-point-response to the comments

Reviewer 3:

The review is in great shape. There are only few areas needs further needs further attention.

Author’s Response: We thank the reviewer for the constructive comments. Our point-by-point response for each comment is provided below:

  1. Please correct the font style in affiliation section.

Author’s Response: We thank the reviewer for the comment. As per the reviewer’s suggestion, we have rectified the font style in the affiliation section.

  1. Figure 2 needs rework on the uniformity and spacing in the design of the arrows.

Author’s Response: We thank the reviewer for the comment. Based on the reviewer suggestion, we have modified the figure 2 and corrected the spacing in the arrows.

  1. The space margins between the lines/paragraphs needs attention.

Author’s Response: We thank the reviewer for the comment. We have checked and modified the space margins throughout the manuscript.

Best Regards,

Dr Alan Prem Kumar
